# Mapping pesticide mixtures to cancer risk at the country scale with spatial exposomics

Jorge Honles [1,2], Juan Pablo Cerapio [2,3], Claudia Monge [2,4,5], Agnès Marchio [2,4,5], Eloy Ruiz [2,6], Ramiro Fernández [2,6], Sandro Casavilca-Zambrano [2,7,8], Juan Contreras-Mancilla [2,7,8], Tatiana Vidaurre [2,9,10], Thomas Condom [11], Swann Zerathe [12], Olivier Dangles [13,14], Éric Deharo [2,15], Javier Herrera-Zuñiga [2,16], Pascal Pineau [2,4,5] & Stéphane Bertani [1,2] ✉

Despite decades of concern over the carcinogenic potential of agricultural pesticides, toxicological studies relying on single endpoints have yet to establish a definitive link between environmental pesticide exposure and cancer in real-world contexts. Here we use an integrative spatial Bayesian framework that merges high-resolution environmental pesticide risk modelling with comprehensive cancer registry data to map pesticide-linked cancer clusters in Peru with unprecedented precision. Our process-based model, encompassing 31 key pesticide active ingredients, together with an innovative stratification of cancer cases by developmental lineage, reveals a robust spatial association between environmental pesticide exposure risk and cancer incidence. In pesticide-associated cancer hotspots, exposomic profiling of liver tissue—a primary target of chemical carcinogens—uncovers a distinct transcriptomic signature of pesticide exposure, implicating a non-genotoxic mode of action that disrupts core regulatory circuitries sustaining cell identity. Collectively, these findings strongly support a mechanistic link between pesticide exposure and cancer, challenging assumptions of human non-carcinogenicity derived from reductionist experimental models. This study redefines the exposome as a lineage-conditioned, mechanistically tractable framework and shows how complex pesticide mixtures can contribute to carcinogenic trajectories, with profound and far-reaching implications for global health policy and socio-ecological equity.

Pesticides pose inherent risks to human health through multifaceted mechanisms[1]. These risks arise from the effects of complex mixtures of active ingredients (AIs) and their degradation products, compounded by interactive effects, non-linear dose–response relationships and variable exposure during the lifespan[2–4]. The pervasive presence of pesticides in food, water and ecosystems renders the characterization of environmental exposures—a key component of the exposome—a daunting challenge[4]. Observational studies often fall short in capturing the complexity of pesticide exposures[5], whereas experimental models tend to oversimplify real-world dynamics[6,7]. Moreover, the tumour-inducing potential of primarily non-genotoxic agents differs markedly between rodent models and humans[8]. Although studies of mixtures yield valuable insights into the health risks from multi-agent exposures, the interplay between individual life histories and socio-environmental context further complicates risk assessment[4,6,7]. Consequently, the real-world carcinogenicity of pesticides remains insufficiently understood, hindering rigorous risk assessments and delaying effective public health interventions. Meeting these challenges requires heuristic

models with enhanced detection and predictive capabilities to identify cancers attributable to pesticide exposure[7].

To address this critical gap, we developed an advanced geospatial modelling approach that offers a transferable blueprint for global environmental health assessments. By integrating high-resolution, process-based environmental models with Bayesian inference that captures population-level heterogeneity in pesticide–cancer risk relationships, our approach maps exposure risk at a fine spatial scale and links statistical associations to mechanistic pathways of carcinogenesis.

Peru, a cradle of plant domestication[9], combines a rich agricultural heritage with acute socio-ecological pressures that push against planetary boundaries[10,11], rendering it a compelling setting for modelling pesticide-related cancer risk. Its diverse geography—from Pacific coastal deserts to Andean highlands and Amazonian rainforests—shapes heterogeneous agricultural practices and pesticide use across a rapidly modernizing sector. Environmental monitoring has revealed alarming levels of contamination[12], while the country's epidemiological transition has brought cancer to the forefront of public health priorities[13,14]. Stark spatial variability in cancer incidence, reflecting socio-ethnic disparities intertwined with territorial dynamics, raises the possibility of environmental factors as potential drivers[13,14]. Moreover, distinctive molecular cancer patterns in Andean–Amazonian regions[15–18], where Indigenous and peasant communities bear disproportionate pesticide exposure burdens[19], further underscore the imperative to elucidate the carcinogenic consequences of pesticide hazards.

## Results

### Process-based environmental pesticide risk mapping

We developed and validated a high-resolution environmental model to assess pesticide risk across Peru and to forecast regions with elevated exposure. Based on pesticide transport and degradation principles, the model computes—on a national scale—the environmental fate of the 31 most commonly used pesticides in the country (Fig. 1 and Extended Data Table 1). To mirror real-world applications, data on AIs from regulatory sources were cross-validated against field surveys of 650 agrochemical retailers (Extended Data Fig. 1). Notably, none of the pesticides included in the model are classified as group 1 carcinogens by the International Agency for Research on Cancer (IARC).

To map environmental pesticide exposure risk, we integrated spatial data on soil properties and monthly resolved hydrometeorological variables (2014–2019) to approximate pesticide transport, degradation and dispersion from application sites to downstream deposition zones (Fig. 1). A hierarchical framework aggregated simulation outputs across all 31 AIs onto a grid with 100 m × 100 m resolution, generating a normalized risk score for each grid cell (scale 0–100). District-level calibration, fine-tuned using 2018 cultivated land cover data, enhanced predictive accuracy by aligning model outputs with local agricultural activity (Extended Data Fig. 2). The model thus captures cumulative, long-term risk from pesticide mixtures by concurrently estimating the environmental behaviour of all 31 AIs, thereby characterizing temporally stable exposure risk surfaces that reflect persistent contamination regimes rather than short-term variability.

By integrating a process-based framework with empirical data, our model reconstructs plausible pesticide exposure scenarios at the district scale (median area = 207.4 km$^2$), capturing the spatial footprint of routine human–environment interactions. To our knowledge, no other system combines national coverage, high spatial resolution and multi-year temporal depth to model chronic exposure risk to a comprehensive panel of key pesticides, making it uniquely suited for spatial epidemiology in Peru.

Nationwide, the model mapped risk levels for 1,793 of 1,874 districts (95.7%; 1,245,451.8 km$^2$), demonstrating its broad geographic coverage despite minor data gaps (Fig. 1 and Extended Data Fig. 2). Zones of moderate and high risk encompassed more than one-third of the national territory, surpassing reference models in both spatial

resolution and predictive precision[20] (Supplementary Fig. 1). Off-site contamination driven by long-range transport extended up to 30–50 km beyond cultivated land (≈1,600 km$^2$). The highest environmental pesticide exposure risks were concentrated in the Andean highlands and slopes, especially along the western flank and southern coastal areas, where limited precipitation exacerbates pesticide accumulation. Conversely, the Amazon basin and northern coast consistently exhibited lower apparent risk levels (Fig. 1 and Extended Data Fig. 2).

To validate real-world model performance, we analysed hair samples from 50 individuals residing in environmentally distinct pesticide risk zones[19]. Biomonitored levels of contaminating AIs and their degradation products exhibited significant spatial autocorrelation (global bivariate Moran's $I$ = 0.42; $z$-score = 1.49; $P$ = 0.044), closely aligning with modelled exposure risk estimates (Extended Data Fig. 3 and Supplementary Table 1). Model sensitivity to climate variability was assessed by comparing two contrasting El Niño–Southern Oscillation (ENSO) phases. Predicted risk significantly increased during the 2015 El Niño episode compared with neutral conditions in 2019 ($P$ = 2.2 × 10$^{-16}$), particularly in the Andes and northern coast (Extended Data Fig. 4). This preliminary finding suggests that climate events may heighten local exposure risk by altering pesticide use patterns[21] and, in turn, the transport and partitioning dynamics that govern their environmental distribution.

### Spatial cancer distribution by cell lineage ontogeny

Building on the pesticide risk model, we mapped the spatial distribution of cancer risk across Peru using data from the Peruvian National Cancer Institute (INEN) registry—the country's most comprehensive source of cancer records[13,14]—for the years 2007 to 2020. Diagnoses were validated by expert pathologists and classified according to the International Classification of Diseases, 10th revision (ICD-10). Key variables—including birth date, sex, residential address at diagnosis, diagnosis date and ICD-10 code—were curated to exclude recurrent cases, duplicates and inconsistencies. Residential histories were verified against the 2007 and 2017 national censuses to ensure a minimum of 5-year residency before diagnosis, yielding a dataset of 158,072 primary cancer cases (C00–C96). Cases were then geocoded using a self-hosted application programming interface (API) and assigned to districts to match the spatial resolution of the environmental pesticide risk model.

Recognizing that cancer development is shaped by developmental lineage programmes[22,23], we stratified the dataset using a molecular framework based on histogenesis and germ layer origin, as proposed by Berman[24] (Fig. 2a). This strategy moves beyond organ-based classifications by grouping cancers according to cell ontogeny, enabling the identification of hallmarks shared among tumours arising from organs with a common developmental lineage. Although often overlooked in cancer classification, lineage dependency has profound implications for environmental epidemiology. At the molecular level, lineage dependency mechanisms are well characterized, with lineage-specific master transcription factors (MTFs) orchestrating the autoregulatory core regulatory circuitries (CRCs) that define cell identity in normal tissues[25] while sustaining cancer cell fitness in a lineage-dependent manner[26–28]. Modulated by extrinsic cues and nuclear hormone receptors[28], these lineage-specific MTFs constitute systemic vulnerabilities that can drive cancer cell transformation in response to environmental stressors[29,30]. This rationale underpinned our adoption of lineage-based stratification in cancer cluster analysis, offering a biologically grounded lens through which to interrogate environmental drivers of tumorigenesis[24].

Within this framework, cancers were stratified into six main categories (Fig. 2a). Endoderm- and ectoderm-derived tumours were classified as either surface ($n$ = 57,890; 36.6%) or parenchymal ($n$ = 32,781; 20.7%). Mesoderm-derived cancers were subdivided into non-mesenchymal ($n$ = 31,881; 20.2%) and mesenchymal ($n$ = 21,518; 13.6%)

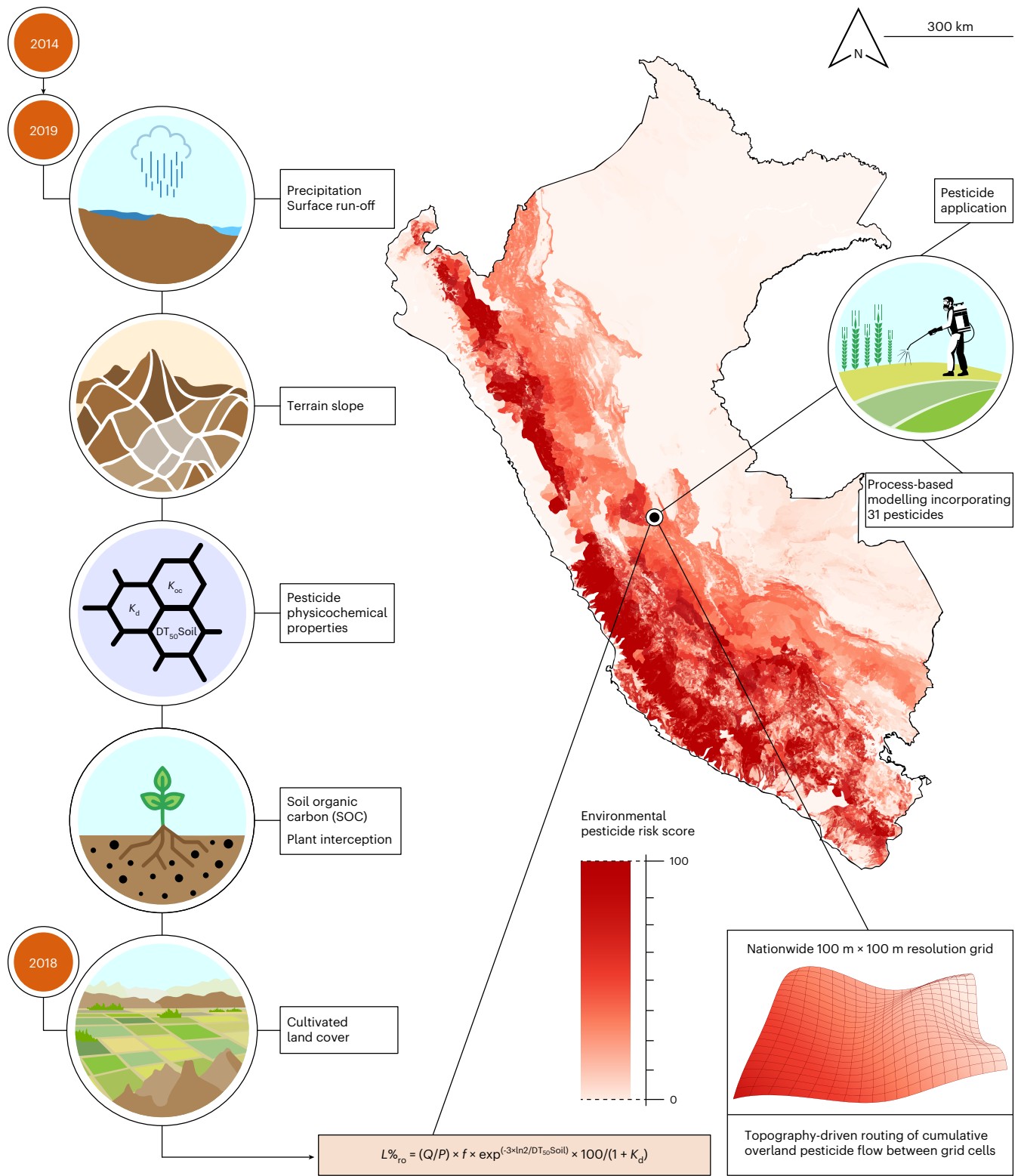

**Fig. 1 | High-resolution environmental model of pesticide risk across Peru, designed to support spatial cancer epidemiology.** This process-based model estimates cumulative pesticide risk across spatial domains and over a 6-year period, both of which are pertinent to chronic human exposure. It integrates spatial data on soil properties and hydrometeorological variables—updated monthly from 2014 to 2019—into a transport framework grounded in surface pesticide mobility and degradation processes, formally represented by an equation described in Methods (see also Supplementary Methods). This framework estimates the environmental fate and dispersal of the 31 most commonly applied pesticides in Peru (Extended Data Table 1). Overland pesticide flow is simulated by routing surface run-off from cell to cell along topography-driven pathways, modelling transport from handheld spraying sites typical of Peruvian agriculture to downstream deposition zones. Outputs across all 31 AIs are aggregated onto a 100 m × 100 m grid to compute a normalized cumulative risk score per cell (scale 0–100), accounting for pesticide mixtures. The map shows the geographical distribution of this score, colour-scaled in red from light (lower risk) to dark (higher risk), highlighting zones of potentially elevated environmental pesticide exposure risk across Peru. Model accuracy is enhanced through calibration using district-level cultivated land cover data (as in Extended Data Fig. 2).

The equation within the figure:

$$L\%_{ro} = (Q/P) \times f \times \exp^{(-3 \times \ln 2/DT_{50}Soil)} \times 100/(1 + K_d)$$

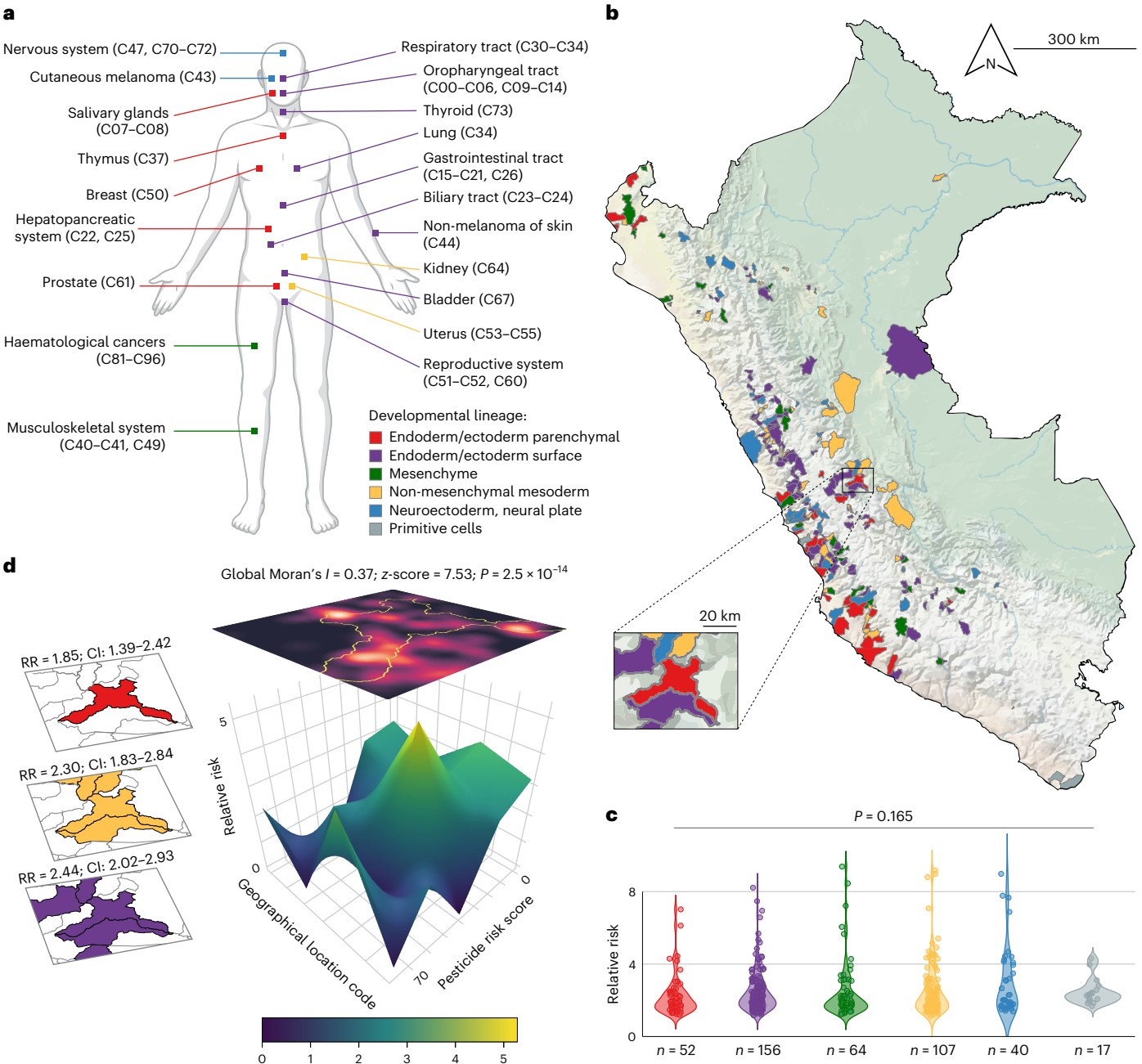

**Fig. 2 | Stratification by cell lineage ontogeny reveals spatial cancer clustering associated with pesticide exposure risk. a**, Developmental mapping of major cancer types across anatomical systems[24], with ICD-10 codes indicating corresponding tumour sites. **b**, Map of Peru showing pesticide-associated cancer hotspots (RR > 1), stratified by developmental lineage. RRs were estimated using a Bayesian geostatistical model fitted with INLA (Supplementary Fig. 2). The inset highlights a hotspot in the Chanchamayo Province (Junín region). **c**, Violin plots showing the distribution of RR estimates across lineages. P value is from a Kruskal–Wallis test. **d**, Left: a spatial overlay identifying a multi-lineage hotspot of pesticide-associated cancers in the Chanchamayo Province. RRs are reported with 95% CIs. Right: a three-dimensional Bayesian posterior mean RR surface for liver cancer (C22.0), depicting spatial variation in modelled cancer

risk as a function of estimated pesticide exposure within the same area. Overlaid is a kernel density estimate heatmap of liver cancer cases, illustrating the spatial alignment of the RR maximum with a geostatistical cluster—confirmed by Moran's $I$ statistic (global Moran's $I$ = 0.37; $z$-score = 7.53; $P$ = 2.5 × 10⁻¹⁴). Locations are slightly jittered to preserve privacy. Colour code (**a–d**) and number of pesticide-associated cancer hotspots (**b** and **c**): endoderm/ectoderm-derived parenchyma, red ($n$ = 52); endoderm/ectoderm-derived surface, violet ($n$ = 156); mesenchyme, green ($n$ = 64); non-mesenchymal mesoderm, yellow ($n$ = 107); neuroectoderm (neural plate), blue ($n$ = 40); primitive cells, grey ($n$ = 17). Illustration in **a** created in BioRender; Bertani, S. https://biorender.com/jo8oap5 (2026).

types. Neuroectodermal tumours, originating from the neural plate, formed a distinct group ($n$ = 7,852; 5%). Primitive cell tumours—including those of germ cell origin ($n$ = 2,076; 1.3%) and those derived from totipotent or multi-potent precursors ($n$ = 194; 0.1%)—were grouped under a 'primitive cells' category. Cancers of undetermined origin (C76–C80; $n$ = 3,558; 2.3%) and tumours with insufficient geostatistical

power—extraembryonic trophoblastic tumours (C58; $n$ = 318; 0.2%) and odontogenic tumours of endodermal/ectodermal origin (C75.2; $n$ = 4; 0.002%)—were excluded from analysis.

To identify cancer clusters based on developmental lineage, we then computed standardized incidence ratios (SIRs) at the district level. Districts with SIR > 1, indicating a higher-than-expected cancer

incidence relative to the general population, were identified as cancer clusters. This approach enabled the generation of high-resolution maps delineating lineage-specific cancer clusters across Peru (Extended Data Fig. 5).

### Lineage-defined cancer clusters associate with pesticide risk

We applied Bayesian inference using the integrated nested Laplace approximation (INLA)—a robust framework for estimating complex spatial models[31]—to forecast regions where a substantial share of the cancer burden is probably attributable to environmental pesticide exposure. Specifically, the method optimizes the geolocation of hotspots, defined as regions where the lower bound of the 95% credible interval (CI) for the relative risk (RR) exceeds 1, indicating a geostatistically significant association between modelled pesticide risk and observed cancer clustering.

Stratifying cancers by developmental lineage improved the INLA model's predictive performance relative to conventional organ-based classifications[32], broadening its applicability, particularly for less common cancers. Lineage-informed modelling identified 436 hotspots across Peru, underpinned by a robust spatial structure and a modelled association between pesticide exposure and cancer incidence (Fig. 2b,c, Extended Data Fig. 6 and Supplementary Fig. 2). RRs within these hotspots ranged from 1.14 to 9.38 (mean ± s.d. = 2.52 ± 1.42), indicating a spatially consistent and epidemiologically relevant increase in cancer risk associated with environmental pesticide exposure. Although RR estimates did not significantly differ across cell lineage groups ($P = 0.165$; Fig. 2c), the distribution and extent of hotspots varied by lineage, suggesting potential lineage-specific susceptibility to local pesticide exposure patterns (Fig. 2b and Extended Data Fig. 6). The most extensive at-risk zones were associated with endodermal and ectodermal epithelial cancers—primarily affecting the gastrointestinal tract (C15–C21), lungs (C34) and skin (C44)—followed by non-mesenchymal, mesoderm-derived malignancies such as those of the female genital organs (C51–C58) and kidney (C64) (Fig. 2a–c and Extended Data Fig. 6).

Geospatial mapping of pesticide-associated cancer risk revealed an intricate interplay among population disparities, land use and Peru's diverse geography. Risk was predominantly concentrated in rural areas experiencing intense anthropogenic pressure. Along the semi-arid Pacific coast, prominent hotspots coincided with zones of modern agriculture on reclaimed and fertilized land[33], notably in Ancash and Piura (north of Lima) and in Ica (south of Lima; Fig. 2b and Extended Data Fig. 1). In the Andes, smaller hotspots emerged in inter-Andean valleys, where steep terrain accelerates pesticide surface run-off[34], probably intensifying local exposure and fostering cancer cluster formation. This pattern was particularly pronounced in the highlands of northern Peru (Amazonas and Cajamarca) and southern Peru (Apurímac, Ayacucho and Cusco; Fig. 2b and Extended Data Fig. 1). Further east, across the Amazon basin, widespread hotspots pertaining to non-mesenchymal, mesoderm-derived cancers spanned the montane and lowland rainforests in the central regions (Huánuco, Junín and Pasco; Fig. 2b and Extended Data Figs. 1 and 6), aligning with deforestation fronts driven by agricultural expansion[35–37] (Extended Data Fig. 7). Including deforestation as a fixed covariate significantly improved the INLA model fit, as evidenced by reductions in the deviance information criterion (DIC = 2,804.24; ΔDIC = 5.73) and the widely applicable information criterion (WAIC = 7,221.95; ΔWAIC = 3.74), reinforcing the spatial association between agricultural pressure and pesticide-associated cancer risk.

Notably, pesticide-associated cancer hotspots in central Peru—amid regions of agricultural deforestation—warrant careful scrutiny, as they overlap with Andean–Amazonian Indigenous and peasant territories, where socio-economic disparities and land-use pressures compound exposure risks[35–37]. Among these, a notable multi-lineage hotspot emerged in Junín (Fig. 2b,d), a region inhabited by Indigenous ethnic groups, including the Ashaninka, Kakinte, Nomatsigenga, Quechua and Yanesha[38] (Extended Data Fig. 8). Spatial autocorrelation analysis based on residential locations uncovered a distinct cluster of liver cancer (C22.0) within this hotspot (Fig. 2d), characterized by strong and statistically significant spatial dependence (global univariate Moran's $I = 0.37$; $z$-score = 7.53; $P = 2.5 \times 10^{-14}$).

### Liver cancer aligns with pesticide exposure hotspots

The endoderm-derived liver is a primary target site for carcinogens in mammals[39–41], making it an ideal organ for validating pesticide-related carcinogenic risks. In Peru, hepatocellular carcinoma (HCC)—the most prevalent liver cancer subtype—disproportionately affects young, non-cirrhotic individuals with Indigenous ancestry[18,42], particularly in central regions such as Junín[15,43]. Tumours in these patients consistently exhibit a molecular profile that diverges from conventional classifications[15,18]. Although hepatitis B virus (HBV) is widely considered the principal risk factor for liver cancer in this population, the frequently occult nature of HBV infection—characterized by low viral DNA levels[44]—along with the distinctive clinical and molecular features of the disease[15,18,42], raises questions about whether HBV alone accounts for these cases.

Geostatistical analyses pinpointed a liver cancer cluster in Junín (Fig. 2d), implicating environmental pesticide exposure as a potential driver. To investigate this link, we performed transcriptomic profiling on paired HCC and non-tumour liver (NTL) samples from 36 non-cirrhotic patients residing in pesticide-associated cancer hotspots[18]. Pesticide-induced hepatotoxicity is primarily mediated by non-genotoxic mechanisms[8,45], although cumulative effects from AI mixtures and their degradation products may elicit modest genotoxic effects[46]. Using the sample enrichment score (SES), an optimized algorithm for single-sample gene set enrichment[47], we detected a gene expression signature indicative of exposure to non-genotoxic carcinogens[48] (Extended Data Table 2). Predominantly present in NTL samples ($P = 5.1 \times 10^{-4}$), this signature points to early exposure preceding malignant transformation (Fig. 3a). We next scored NTL samples for gene expression signatures linked to established risk factors for liver cancer[49], including HBV infection, high alcohol consumption, steatosis and foodborne genotoxins such as aflatoxin $B_1$ (Extended Data Table 2). Statistical analyses confirmed preferential tumour induction by non-genotoxic carcinogens (Friedman test; $Q = 112.6$, $P = 2 \times 10^{-23}$), a pattern visually apparent in Fig. 3b (see also Supplementary Table 2 and Supplementary Fig. 3). To determine whether this non-genotoxic signature was specific to the pesticide-associated cancer hotspots in Peru or reflected broader environmental exposure patterns, we compared the Peruvian dataset with transcriptome datasets from France, Taiwan and Turkey. The results confirmed that the signature was unique to Peruvian patients, with no analogous signal detected in any other cohort ($P = 6.8 \times 10^{-15}$; Fig. 3a,b).

Prompted by geospatial cancer cluster analyses stratified by developmental lineage, we examined the expression of nine lineage-specific MTFs that orchestrate CRCs and drive endoderm-derived hepatobiliary carcinogenesis[26] (Extended Data Table 2). Remarkably, NTL samples from Peruvian patients exhibited significantly higher SES for these MTFs than those from France, Taiwan and Turkey ($P = 5.7 \times 10^{-16}$; Fig. 3c and Supplementary Fig. 4), indicative of CRC disruption. These findings suggest early systemic disruption of the MTF-driven autoregulatory loop sustaining CRC-dependent hepatocyte identity[50], occurring before malignant transformation. This preneoplastic alteration is consistent with a field-effect-like phenomenon in histologically normal liver tissue[42], whereby some hepatocytes, despite retaining functionality, probably enter an unstable steady state within a saddle-node epigenetic landscape[51] (Fig. 3d). While this epigenetic landscape is shaped by multiple layers of regulation, our data indicate that aberrant DNA methylation[18]—triggered by pesticide-induced CRC disruption[52] and maintained in a lineage-specific manner by the polycomb repressive complex 2 (PRC2)[53,54], a key effector in Peruvian liver

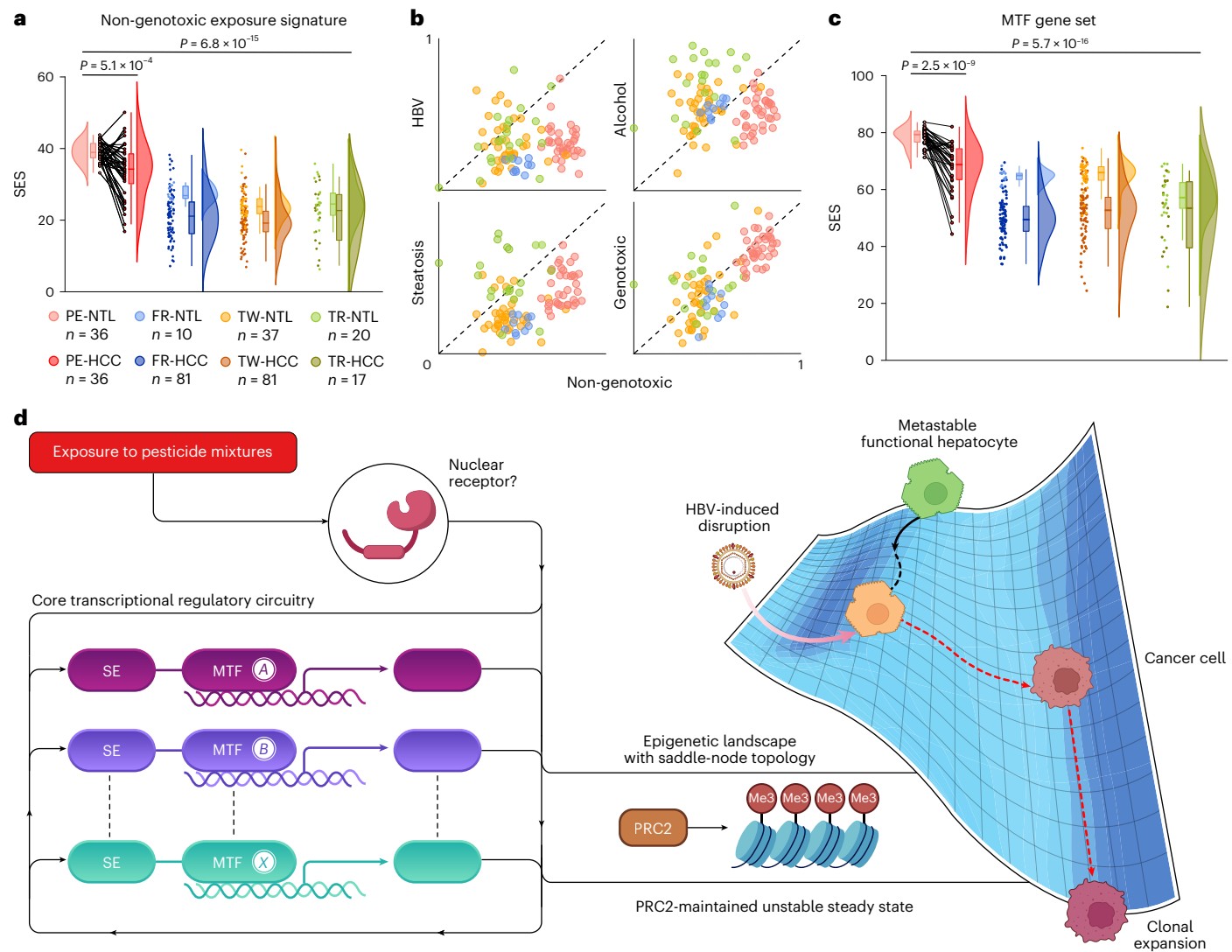

**Fig. 3 | Transcriptomic evidence of non-genotoxic carcinogenesis and early lineage-specific regulatory disruption in pesticide-associated liver cancer. a**, Raincloud plots of SES for a curated non-genotoxic exposure signature in HCC and NTL samples[48] (Extended Data Table 2). **b**, Scatter plots of min−max scaled SES values in NTL samples, comparing the non-genotoxic signature with transcriptomic signatures linked to major HCC risk factors. A dashed identity line ($x = y$) highlights deviations between signatures. **c**, Raincloud plots of SES for a curated gene set of lineage-specific MTFs implicated in hepatobiliary carcinogenesis[26]. In **a**−**c**, Peruvian samples (PE; red; $n = 36$ NTL and $n = 36$ HCC) are compared with cohorts from France (FR; blue; $n = 10$ NTL and $n = 81$ HCC), Taiwan (TW; orange−brown; $n = 37$ NTL and $n = 81$ HCC) and Turkey (TR; green; $n = 20$ NTL and $n = 17$ HCC); lighter and darker shades denote NTL and HCC, respectively. Raincloud plots in **a** and **c** combine box plots with kernel density estimates. Box plots indicate the median (centre line), interquartile range (IQR, box) and 1.5 × IQR bounds (whiskers), with outliers beyond the whiskers

plotted individually as points. Peruvian samples are matched pairwise. $P$ values are from Kruskal–Wallis tests followed by Dunn's multiple-comparison test with correction and from two-sided paired Wilcoxon tests. **d**, Schematic summarizing the proposed mechanism by which pesticide mixtures induce early systemic disruption of the CRC formed by super-enhancers (SEs) and MTFs—an autoregulatory loop essential for sustaining hepatocyte identity[50]. This disruption propels cells into an unstable, preneoplastic steady state situated within a saddle-node epigenetic landscape[51]—a departure from Waddington's model. Characterized by aberrant DNA methylation and maintained by PRC2[18], this metastable state constitutes a transformation-prone, epigenetically entrapped configuration that destabilizes cell-fate commitment. Dysregulation of PRC2, frequent in HBV infection[55], may trigger the collapse of this precarious equilibrium, culminating in malignant transformation. Schematic in **d** adapted with permission from ref. 51, Elsevier; icons in **d** created in BioRender; Bertani, S. https://biorender.com/3v25ddg (2026).

cancer[18]—compromises cellular stability, thereby undermining proper cell-fate commitment and predisposing the tissue to malignant transformation. Subsequent dysregulation of PRC2 through its SUZ12 subunit during HBV infection[55] may breach this unstable epigenetic milieu, acting as a tipping point that precipitates malignant transformation.

## Discussion

This study links pesticide exposure to cancer risk in a nationwide cohort, revealing mechanistic pathways of environmentally driven carcinogenesis, with important implications for global health policy.

Our advance stems from integrating a state-of-the-art, process-based environmental risk model with Bayesian spatial epidemiology and transcriptomic validation. This strategy enabled high-resolution mapping of cumulative environmental exposure risk to pesticide mixtures—unprecedented in scope—with each AI individually deemed non-carcinogenic[56]. The resulting risk surfaces were linked to spatial patterns of cancer incidence. By stratifying comprehensive cancer registry data according to developmental lineage, we redefined cancer clustering within a coherent ontogenetic framework, uncovering hallmark patterns often obscured by conventional organ-based

classifications[24]. This biologically anchored approach affords a nuanced assessment of pesticide-associated cancer risk, exemplified by the identification of an atypical molecular subtype of liver cancer in Andean–Amazonian Indigenous patients[18], whose aetiology had remained elusive.

Our findings indicate that pesticide exposure may contribute to the risk of liver cancer in this population. At the molecular level, chronic exposure to non-genotoxic pesticide mixtures—even at or below individual experimental no-observed-adverse-effect levels[3]—disrupts lineage-dependent transcriptional programmes and fosters cancer development. Transcriptomic analyses show that such exposures perturb the expression of lineage-specific MTFs in normal tissue, destabilizing the regulatory networks that sustain cell identity[25,50] and undermining the epigenetic safeguards that enforce cell-fate commitment[52]. This aberrant transcriptional state probably increases cellular vulnerability to extrinsic cues—whether environmental, metabolic, infectious or inflammatory[30,49]—that may act as tipping points for malignant transformation, most notably through mild or even occult HBV infection in Peruvian patients with liver cancer[44]. These findings raise a critical question: do disruptions in lineage-specific MTF expression accumulate over time, or are they imprinted during early development[57], in line with the DOHaD (developmental origins of health and disease) hypothesis[58]? While our data implicate the liver—a primary target of chemical carcinogens[39–41]—as a sentinel site of lineage-specific disruption of MTF activity, it remains undetermined whether analogous mechanisms occur in other exposed tissues, which were not assessed here. Further investigation of additional organ sites identified through geospatial clustering is warranted to determine whether this lineage-dependent susceptibility reflects a broader systemic response to pesticide mixtures.

Beyond its molecular insights, our study reveals pressing socio-environmental challenges. In regions where intensive agriculture, unsustainable land management and limited healthcare coalesce[10,13], the dispersal of pesticides not only undermines ecological resilience[59] but also exacerbates enduring health inequalities. Geospatial modelling reveals that high-risk zones for pesticide-associated cancer are disproportionately concentrated in rural areas experiencing intense anthropogenic pressure. This pattern underscores the intricate interplay between environmental degradation, land-use dynamics and socio-economic marginalization, placing underserved communities at greatest risk and reinforcing the urgent need for targeted policy interventions[35]. Moreover, preliminary evidence suggests that climate phenomena such as ENSO shape land-use patterns and influence pesticide dispersion[21], thereby amplifying local exposure risk. As global warming advances, these dynamics are likely to intensify, further aggravating the burden of environmentally driven cancers among exposed populations.

Despite the robustness of our integrative approach, certain limitations merit consideration. First, individual pesticide exposures were not directly measured but inferred from population-level spatial proxies, introducing uncertainty regarding their timing, intensity and chemical composition, as well as precluding direct attribution of cancer onset at the individual level. Second, residual confounding from environmental, socio-economic or lifestyle factors cannot be entirely excluded; nevertheless, the convergence of geospatial, aggregated epidemiological and molecular evidence strongly supports the biological plausibility of our findings. Future studies incorporating large-scale biomonitoring, individual-level exposure assessment, multi-tissue analyses and longitudinal molecular profiling will be essential to strengthen causal inference, evaluate generalizability and establish direct links between pesticide exposure and cancer incidence, building on the framework established herein.

Taken together, this work establishes a unified probabilistic model to quantify cancer risk across diverse tumour types and real-world pesticide exposure scenarios. By transcending the limitations of reductionist toxicology[7], our approach offers a scalable and transferable framework for global environmental health assessment. Crucially, our findings underscore the imperative of embedding socio-ecological equity within regulatory policy—an essential step towards mitigating ecological harm and protecting vulnerable populations from environmentally driven cancers.

## Methods

### Ethics and data protection compliance
The study complies with Peru's Personal Data Protection Law 29733 (Regulation PDPL 016-2024-JUS). The INEN Cancer Registry is authorized by the Peruvian National Authority for the Protection of Personal Data under registration number RNPDP-EP 4794. Ethical approvals were granted by the INEN Institutional Review Board under reference numbers 113-2014-CIE/INEN, 407-2016-CIE/INEN and 049-2024-DICON/INEN. All procedures adhered to the ethical principles of the WMA Declaration of Taipei on research involving health databases, big data and biobanks.

### Survey of agrochemical retailers
A survey, described in detail elsewhere[60], was conducted in 2020 to identify the pesticides most commonly used in Peru and to detect any continued use of obsolete or banned products[61], as mandated by the Peruvian National Agricultural Health Service (SENASA). A total of 650 agrochemical retailers—both formal (as listed in the Integrated Agricultural Input Management System (SIGIA); https://servicios.senasa.gob.pe/SIGIAWeb/sigia_consulta_empresa.html; accessed 24 October 2020) and informal (including itinerant and market-based vendors)—were randomly sampled across five agriculturally diverse regions: Ayacucho, Huancavelica, Huánuco, Junín and Pasco (Extended Data Fig. 1).

A pretested, semistructured questionnaire[62] was used to document commercial pesticide names and application frequencies. Interviews were conducted face to face by a native Spanish-speaking agronomist, with responses recorded electronically using KoboToolbox (v.2.020.25). The response rate reached 98.9%, ensuring representative coverage of local pesticide practices. Commercial names were standardized using OpenRefine (v.3.4.1), and unmatched entries were excluded. AIs and their Chemical Abstracts Service (CAS) registry numbers were retrieved from the PubChem database (https://pubchem.ncbi.nlm.nih.gov/) using OpenRefine's reconciliation function and cross-validated against national regulatory records to confirm that the final list of AIs accurately reflects pesticide mixtures present in environmental contexts.

### Pesticide inventory for environmental modelling
A total of 31 AIs—comprising 19 insecticides, 7 fungicides and 5 herbicides (Extended Data Table 1)—were retained for environmental modelling. All, with the exception of monocrotophos (banned by SENASA in 2004), were authorized for use in Peru during the 6-year modelled period (2014–2019). None was classified as carcinogenic to humans (group 1) in the IARC Monographs (https://monographs.iarc.who.int/list-of-classifications; accessed 28 May 2025), nor deemed extremely hazardous (Ia) under the World Health Organization (WHO) classification of pesticides by hazard[63].

### Process-based environmental pesticide risk modelling
The environmental pesticide risk model was built on a grid of 124,545,177 cells (100 m × 100 m; 0.01 km² each), covering 96.7% of Peru's 1,287,072.5-km² national territory. Environmental variables integrated into the model included monthly precipitation and surface run-off (millimetres per month; 2014–2019; PISCO_HyM_GR2M v.1.1), terrain slope (degrees) derived from elevation data (metres above sea level; 2014; EarthEnv-DEM90 v.1) and soil organic carbon (SOC, %) calculated from the Global Soil Organic Carbon Map v.1.5

(2018; GSOCmap-FAO) and topsoil bulk density from the Harmonized World Soil Database v.1.2 (2014; T_REF_BULK v.1.0). All spatial layers were standardized to Universal Transverse Mercator (UTM) zone 18S (WGS84, EPSG:32718), converted to raster format and resampled to 100 m × 100 m resolution (≈3.2 arcsec at the equator) using bilinear interpolation to ensure cross-dataset compatibility.

Physicochemical properties of AIs were sourced from the Chemical Entities of Biological Interest (ChEBI) database (https://www.ebi.ac.uk/chebi/), the Pesticide Properties Database (PPDB; https://sitem.herts.ac.uk/aeru/ppdb/en/) and PubChem. Environmental pesticide risk for each AI was modelled using a lumped-parameter expression developed by the Organization for Economic Co-operation and Development (OECD) expert group[64]. This equation approximates key transport and dissipation processes—run-off mobilization, sorption and degradation—as follows:

$$L\%_{ro} = \left(\frac{Q}{P}\right) \times f \times \exp^{(-3 \times \ln 2/DT_{50}Soil)} \times \frac{100}{(1 + K_d)}$$

where $L\%_{ro}$ is the estimated percentage of applied pesticide available for run-off in a dissolved form; $\frac{Q}{P}$ is the run-off to precipitation ratio, calculated monthly for each grid cell; $f$ is a correction factor accounting for terrain slope, plant interception and buffer zones (Supplementary Methods); $DT_{50}Soil$ is the soil dissipation half-life; and $K_d$ is the soil–water distribution coefficient, derived from the organic carbon distribution coefficient ($K_{oc}$) and local SOC content (%OC) (as in Extended Data Table 1).

A total of 2,232 raster layers, representing monthly outputs for 31 pesticides over 72 months (2014–2019), were averaged to produce the final environmental risk surface. Risk scores were normalized to a 0–100 scale to enable both spatial and intercompound comparisons. Model calibration was performed using the 2018 cultivated land cover (km²), derived from satellite imagery by the Peruvian Ministry of Agriculture and Irrigation (MINAGRI)[65]. Zonal statistics were computed using the 2021 district boundaries from the Spatial Data Infrastructure of Peru (IDEP). Normalized raster layers were overlaid with district polygons, and summary metrics (mean, median, minimum and maximum) were calculated across all raster cells within each district. All computations were performed using QGIS v.3.28 LTR and R v.4.4.1 (packages: sf v.1.0-10, sp v.1.5-1 and terra v.1.8-70). Database access details are provided in Supplementary Table 3. Hard data underlying the environmental pesticide exposure risk map are publicly available at https://doi.org/10.6084/m9.figshare.29728463.

### Climatic phase determination
Climatic phases were characterized based on ENSO variability, defined by the 3-month running average Oceanic Niño Index (ONI) derived from sea surface temperature anomalies in the Niño 1 + 2 region (0–10° S, 90–80° W). Monthly sea surface temperature data were obtained from the Optimum Interpolation Sea Surface Temperature (OISST) v.2.1 dataset (1991–2020 base period), available from the Climate Prediction Center (CPC; https://www.cpc.ncep.noaa.gov/data/indices/sstoi.indices). For sensitivity analyses, the 2015 'Godzilla' El Niño event[66] and the 2019 neutral phase were selected as contrasting climatic conditions within the modelled period (2014–2019).

### Hair biomonitoring and multi-residue chemical analysis
Hair sampling, described in detail elsewhere[19], was performed in 2020 among 50 randomly selected adults from four Peruvian regions—Huancavelica, Ica, Junín and Lima (Extended Data Fig. 3)—with equal representation by sex (1:1) and residence (25 rural, 25 urban). Sampling locations were georeferenced using an eTrex 20 GPS device (Garmin). Hair was collected according to the manufacturer's (Kudzu Science) protocol. A 5-mm circular guide was used to isolate a random section, which was cut close to the scalp using sterilized scissors (70% ethanol).

Hair samples were wrapped in aluminium foil, labelled at the root ends and stored dry at 4 °C for 2 weeks before analysis. A hair growth rate of 1 cm per month was assumed[67]; the proximal 0.5 cm (embedded in the scalp) was excluded from analysis, as it is inaccessible for sampling.

Levels of 67 AIs and degradation products were quantified in hair (pg mg⁻¹)[19], spanning 19 pesticides incorporated into the environmental exposure risk model, as detailed in Supplementary Methods and Supplementary Table 1.

### Stratification of cancer registry data by developmental lineage
The developmental lineage classification of neoplasms, originally developed by Berman[24,68,69], is a validated taxonomy that organizes neoplasms by embryonic origin within a structured hierarchy of 39 classifiers across seven levels. To balance biological coherence with statistical robustness, we stratified data from the INEN Cancer Registry (2007–2020) to the fifth hierarchical level. The resulting schema was as follows:

```
neoplasms
    embryonic
        primitive
            primitive_differentiating
            germ_cell
        non_primitive
            endoderm_or_ectoderm
                endoderm_or_ectoderm_surface
                endoderm_or_ectoderm_parenchymal
                odontogenic_epithelium
            mesoderm
                mesenchyme
                non_mesenchymal_mesoderm
            neuroectoderm_neural_plate
    extra_embryonic
        trophoblastic
```

The full developmental lineage classification with taxonomy, encoded in eXtensible Markup Language (XML) format and comprising approximately 55,000 hierarchically organized neoplasia terms, was sourced from Berman's seminal article[24]. A custom Python parser extracted the lineage mapping and exported it as a CSV file to streamline processing. Neoplasia terms were linked to ICD-10 Clinical Modification (ICD-10-CM) codes using a predefined crosswalk to concepts in the US National Cancer Institute Thesaurus (NCIt_Maps_To_ICD10CM v.25.03e; https://evsexplore.semantics.cancer.gov/evsexplore/welcome). Discrepancies were systematically logged and curated by experts, with constraint-based mapping enforcing strict one-to-one and hierarchical consistency. The resulting ICD-10-mapped developmental lineage classification was imported into a PostgreSQL relational database, with SQL constraints applied to enforce referential integrity.

Cancer registry data—including birth date, sex, residential location at diagnosis, diagnosis date and ICD-10 code—were extracted from the INEN Hospital Information System (SISINEN v.2.0), hosted on Oracle Database v.19c (Oracle Corporation) and exported in XML format. These data were processed in Python, combining automated cleaning with manual validation to ensure stringent curation. Residential histories were consolidated to confirm at least 5 years of residence and cross-validated against Peru's 2007 and 2017 national censuses (operated by the Peruvian National Institute of Statistics and Informatics (INEI)). They were then geocoded using a self-hosted instance of the Nominatim API v.4.5.0 with OpenStreetMap data (https://www.openstreetmap.org/; accessed 24 September 2024). The final curated dataset, comprising 158,072 cases of malignant neoplasms (ICD-10-CM codes C00–C96), was imported into the PostgreSQL database. ICD-10-CM codes served as foreign keys, ensuring hierarchical integrity and enabling the integration of the geocoded dataset within the

developmental lineage framework. The classified dataset—including ICD-10 codes, case counts and assigned developmental lineages—is publicly available at https://doi.org/10.6084/m9.figshare.29728463.

## SIR mapping

Standardized incidence estimation was performed and aggregated by district. Key attributes—including cancer categories, geocoded case counts, population denominators from the 2007 and 2017 national censuses (INEI), and district-level identifiers from 2021 (Geographic Location Code (UBIGEO) v.2.0)—were integrated within a PostgreSQL relational database. Stratification was applied across six strata defined by sex (female, male) and age (0–39, 40–59 and ≥60 years) groups. Expected case counts were estimated by indirect standardization using the following formula:

$$E_i = \sum_{j=1}^{m} r_j^{(s)} \eta_j$$

where $r_j^{(s)}$ is the disease rate in stratum $j$ of the standard population, and $\eta_j$ is the population in stratum $j$ of district $i$. SIRs were then calculated as

$$\text{SIR}_i = Y_i / E_i$$

SIR mapping was performed using QGIS v.3.28 LTR and R v.4.4.1 (packages: SpatialEpi v.1.2.8 and terra v.1.8-70). Database access details are provided in Supplementary Table 3. District-level SIR values with UBIGEO codes are publicly available at https://doi.org/10.6084/m9.figshare.29728463.

## Geospatial Bayesian modelling

To estimate pesticide-associated cancer risk, we implemented a latent Gaussian model using INLA, specified as

$$y_i | \theta_i \sim \text{Po}\left(E_i \times \theta_i\right), i = 1, \dots, n$$

where $E_i$ is the expected count and $\theta_i$ is the RR in district $i$. Observed cases were assumed to follow a Poisson distribution. RR was modelled as

$$\log(\theta_i) = \beta_0 + \beta_1 P_i + u_i + v_i$$

where $\beta_0$ is the intercept, $\beta_1$ the coefficient for environmental pesticide exposure, $P_i$ and $u_i$ the spatially structured effects and $v_i$ the unstructured effect. Spatial dependence was defined using an adjacency matrix, and the logarithm of population size was included as an offset. Posterior means of the linear predictor were interpreted as RRs, with 95% CIs defined by the 2.5th and 97.5th percentiles (Supplementary Fig. 2). Spatial outputs were aggregated and visualized by developmental lineage. District-level RR values with 95% CIs and UBIGEO codes are publicly available at https://doi.org/10.6084/m9.figshare.29728463.

Following the initial model specification, we incorporated data on emerging forest loss clusters[70]—retrieved from the Global Forest Watch (GFW) platform (2002–2023; https://www.globalforestwatch.org/; accessed 23 January 2025)—as fixed covariates. To evaluate their impact on model fit and cancer risk estimates, we conducted a sensitivity analysis comparing Poisson and zero-inflated Poisson likelihoods and performed null-model simulations to assess baseline performance. Competing models were compared using DIC and WAIC. All analyses were conducted in R v.4.4.1 (packages: INLA v.25.3.24, sf v.1.0-19 and terra v.1.8-70).

## Transcriptomic profiling

Transcriptomic analyses were conducted on paired HCC and adjacent NTL tissues from 39 surgical specimens collected from Peruvian patients[18]. Raw expression data were retrieved from the Gene Expression Omnibus (GEO) database (https://www.ncbi.nlm.nih.gov/geo/). Three HCC–NTL pairs (patients PE091, PE104 and PE201) were excluded because they originated from individuals residing outside pesticide-associated cancer hotspots, yielding a final dataset of 36 matched pairs. All patients underwent anatomical liver resection at INEN between 2006 and 2016, with no prior chemotherapy or radiotherapy[71,72]. NTL samples were collected from tumour-free resection margins at least 1 cm wide. Histopathological review confirmed the HCC diagnosis and negative margins, while tissue characterization encompassed steatosis grade, fibrosis stage and key genetic and viral features to support risk factor inference (Supplementary Methods and Supplementary Table 2).

Briefly, 50 mg of HCC or NTL tissue was collected following resection by expert hepatobiliary pathologists, flash-frozen in liquid nitrogen and stored at −150 °C in the National Tumour Bank at INEN before analysis. Frozen tissues were pulverized under liquid nitrogen and digested for 8 h at 37 °C in RNase-free lysis buffer containing SDS and proteinase K. Total RNA was extracted using TRI Reagent (Sigma-Aldrich) and homogenized with the Lysing Matrix D system (MP Biomedicals). RNA yield was quantified using the Qubit RNA Broad-Range Assay Kit (Invitrogen, Thermo Fisher Scientific), and RNA integrity was assessed with the RNA 6000 Nano LabChip Kit on a 2100 Bioanalyzer instrument (Agilent Technologies). Only samples with an RNA integrity number greater than 7 were retained. Transcriptome profiling was then performed using the GeneChip Human Transcriptome Array 2.0 and the GeneChip WT PLUS Reagent Kit (Applied Biosystems, Thermo Fisher Scientific), following the manufacturer's protocol.

For cross-population comparisons, transcriptome datasets from HCC and NTL tissues of patients from France ($n = 81$ HCC; $n = 10$ NTL), Taiwan ($n = 81$ HCC; $n = 37$ NTL) and Turkey ($n = 17$ HCC; $n = 20$ NTL) were obtained from GEO; all were profiled on the same GeneChip platform as the Peruvian cohort. Data were normalized using the robust multi-array average algorithm, batch-corrected with the ComBat function from the sva package (v.3.54.0, Bioconductor v.3.20; https://www.bioconductor.org/) and collapsed using Collapse Microarray (Fred's Softwares; https://sites.google.com/site/fredsoftwares/home).

Single-sample gene set enrichments were computed as SES values[47] using the AutoCompare SES algorithm (Fred's Softwares), following the recommended workflow for direct sample-to-sample comparison. Min–max scaling was applied to the SES to ensure comparability across gene expression signatures. Gene sets were curated from the literature to capture transcriptomic responses[73] to liver-specific exposures[48,74–77] and lineage-specific MTFs implicated in endoderm-derived hepatobiliary carcinogenesis[26], as detailed in Extended Data Table 2. SES values are publicly available at https://doi.org/10.6084/m9.figshare.29728463.

## Statistical analysis and visualization

The specific statistical tests used are indicated in the main text or in the figure legends. Spatial autocorrelation was assessed using global and local univariate or bivariate Moran's $I$ statistics, as appropriate for the data type, to detect overall clustering and localized patterns, with significance evaluated by Monte Carlo permutation testing (10,000 iterations). Raster datasets representing environmental pesticide risk were compared between the 2015 and 2019 ENSO phases using a paired Student's $t$ test. Group comparisons were conducted using the Wilcoxon signed-rank test for paired data and the Kruskal–Wallis or Friedman tests for multiple groups, followed by Dunn's post hoc test with Bonferroni correction. All statistical tests were performed at a 0.05 significance level, using two-sided procedures where applicable.

Statistical analyses and data visualization were conducted using Bioconductor (v.3.20), Excel (v.16.16.27), Fred's Softwares, GraphPad Prism (v.10.4.1), Mitomaster (v.Beta 1), Nominatim (v.4.5.0), PostgreSQL (v.16.3), Python (v.3.11.3), QGIS (v.3.28 LTR), QuantaSoft (v.1.7) and R (v.4.4.1). R packages included dplyr (v.1.1.4), gghalves (v.0.1.4),

ggplot2 (v.3.5.2), ggrain (v.1.0.2), INLA (v.25.3.24), Plotly (v.4.10.4), raincloudplots (v.0.2.0), sf (v.1.0-10), sp (v.1.5-1), SpatialEpi (v.1.2.8), spdep (v.1.3-10), sva (v.3.54.0), terra (v.1.8-70) and tmap (v.4.0). KoboToolbox (v.2.020.25) and OpenRefine (v.3.4.1) were used for data collection and cleaning, respectively. Figures were assembled using Adobe Illustrator (v.29.0). Schematic diagrams in Figs. 2a and 3d were created in Adobe Illustrator and BioRender (https://BioRender.com). Administrative boundaries in Figs. 1 and 2, Extended Data Figs. 1–8 and Supplementary Figs. 1 and 2 were derived from INEI shapefiles (Supplementary Table 3) and are shown for analytical purposes only, without implying official designation.

### Reporting summary

Further information on research design is available in the Nature Portfolio Reporting Summary linked to this article.

## Data availability

The datasets generated and analysed during the current study are available in the figshare repository at https://doi.org/10.6084/m9.figshare.29728463 (ref. 78), except for individual-level geolocation data from the GEO database and the INEN Cancer Registry, which are not publicly available to protect privacy and comply with data protection regulations. The transcriptomic data analysed are publicly available in GEO under accession numbers GSE111580 and GSE136247 (Peru). Comparative datasets were obtained from GEO under accession numbers GSE17548 (Turkey), GSE45436 (Taiwan) and GSE62232 (France). No new individual-level data were collected, and informed consent was not required.

## Code availability

The code used for data analysis and modelling is available at https://doi.org/10.6084/m9.figshare.29728463 (ref. 78).

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

## Acknowledgements

We thank all individuals whose participation made this study possible. We are grateful to F. Colas, J.-C. Espinoza, J.-J. Fournié, C. Junquas, M. Kelly-Irving, L. Orlando and A. Valentin for their comments, as well as to the staff of INEI and the Peruvian National Tumour Bank at INEN for their support. This work was funded by ITMO Cancer of the French National Alliance for Life Sciences and Health (Aviesan) and the French National Cancer Institute (INCa), with funds administered by the French National Institute of Health and Medical Research (INSERM) under grant agreement 21CD025-00 (S.B.). Additional support was provided by the French National Agency for Research on AIDS and Viral Hepatitis (ANRS) Emerging Infectious Diseases, with funds administered by INSERM under grant agreement ECTZ158014 (S.B.). The funding bodies had no role in the study design, data collection, analysis and interpretation of data, or the writing of this article.

## Author contributions

Designed, conceived and coordinated the study: S.B. Provided access to multi-domain data: T.C., S.Z., O.D., J.H.-Z. and S.B. Provided access to cancer registry data: S.C.-Z., T.V. and S.B. Conducted field research: J.H., C.M. and S.B., with input from O.D., E.D. and P.P. Provided samples, reagents and materials: E.R., R.F., S.C.-Z., E.D., P.P. and S.B. Modelled environmental pesticide risk: J.H., with input from T.C., S.Z., J.H.-Z. and S.B. Performed geostatistical analyses: J.H., with input from S.B. Performed dry laboratory analyses: J.H. and J.P.C., with input from S.C.-Z. and S.B. Performed wet laboratory work: J.P.C., C.M. and A.M., with input from P.P. and S.B. Analysed transcriptomic data: J.P.C., J.C.-M., P.P. and S.B. Interpreted the data: J.H. and S.B., with input from P.P. Wrote the paper: S.B., with input from all coauthors. Prepared the Supplementary Information: J.H. and S.B., with input from all coauthors. All coauthors approved the final version.

## Competing interests

The authors declare no competing interests.

## Additional information

**Extended data** is available for this paper at https://doi.org/10.1038/s44360-026-00087-0.

**Correspondence and requests for materials** should be addressed to Stéphane Bertani.

¹Unité Pharmacochimie et Biologie pour le Développement (UMR 152 PHARMA-DEV), IRD, Université de Toulouse, Toulouse, France. ²International Joint Laboratory for Molecular Anthropology of Cancer and Oncogenic Viruses (LOAM), IRD, Instituto Nacional de Enfermedades Neoplásicas, Lima, Peru. ³Centre de Recherches en Cancérologie de Toulouse (UMR 1037 CRCT), CNRS, INSERM, Université de Toulouse, Toulouse, France. ⁴Unité Organisation Nucléaire et Oncogenèse, Institut Pasteur, INSERM (U 993), Université Paris Cité, Paris, France. ⁵Unité Virus et Stress Cellulaire, Institut Pasteur, Université Paris Cité, Paris, France. ⁶Departamento de Cirugía en Abdomen, Instituto Nacional de Enfermedades Neoplásicas, Lima, Peru. ⁷Departamento de Patología, Instituto Nacional de Enfermedades Neoplásicas, Lima, Peru. ⁸Banco Nacional de Tumores, Instituto Nacional de Enfermedades Neoplásicas, Lima, Peru. ⁹Departamento de Oncología Médica, Instituto Nacional de Enfermedades Neoplásicas, Lima, Peru. ¹⁰Facultad de Medicina, Universidad Peruana Cayetano Heredia, Lima, Peru. ¹¹Institut des Géosciences de l'Environnement (UMR 5001 IGE), CNRS, INRAE, IRD, Université Grenoble Alpes, Grenoble, France. ¹²Institut des Sciences de la Terre (UMR 5275 ISTerre), CNRS, IRD, Université Grenoble Alpes, Grenoble, France. ¹³Centre d'Écologie Fonctionnelle et Évolutive (UMR 5175 CEFE), CNRS, EPHE, IRD, Université de Montpellier, Montpellier, France. ¹⁴Centro en Ciencias de la Sostenibilidad (WasiLab), IRD, Pontificia Universidad Católica del Ecuador, Quito, Ecuador. ¹⁵HKU-Pasteur Research Pole, Li Ka Shing Faculty of Medicine, The University of Hong Kong, Hong Kong, China. ¹⁶Laboratoire d'Économie de Dauphine (UMR 8007-260 LEDa), CNRS, IRD, Université Paris Dauphine-PSL, Paris, France. ✉e-mail: stephane.bertani@ird.fr

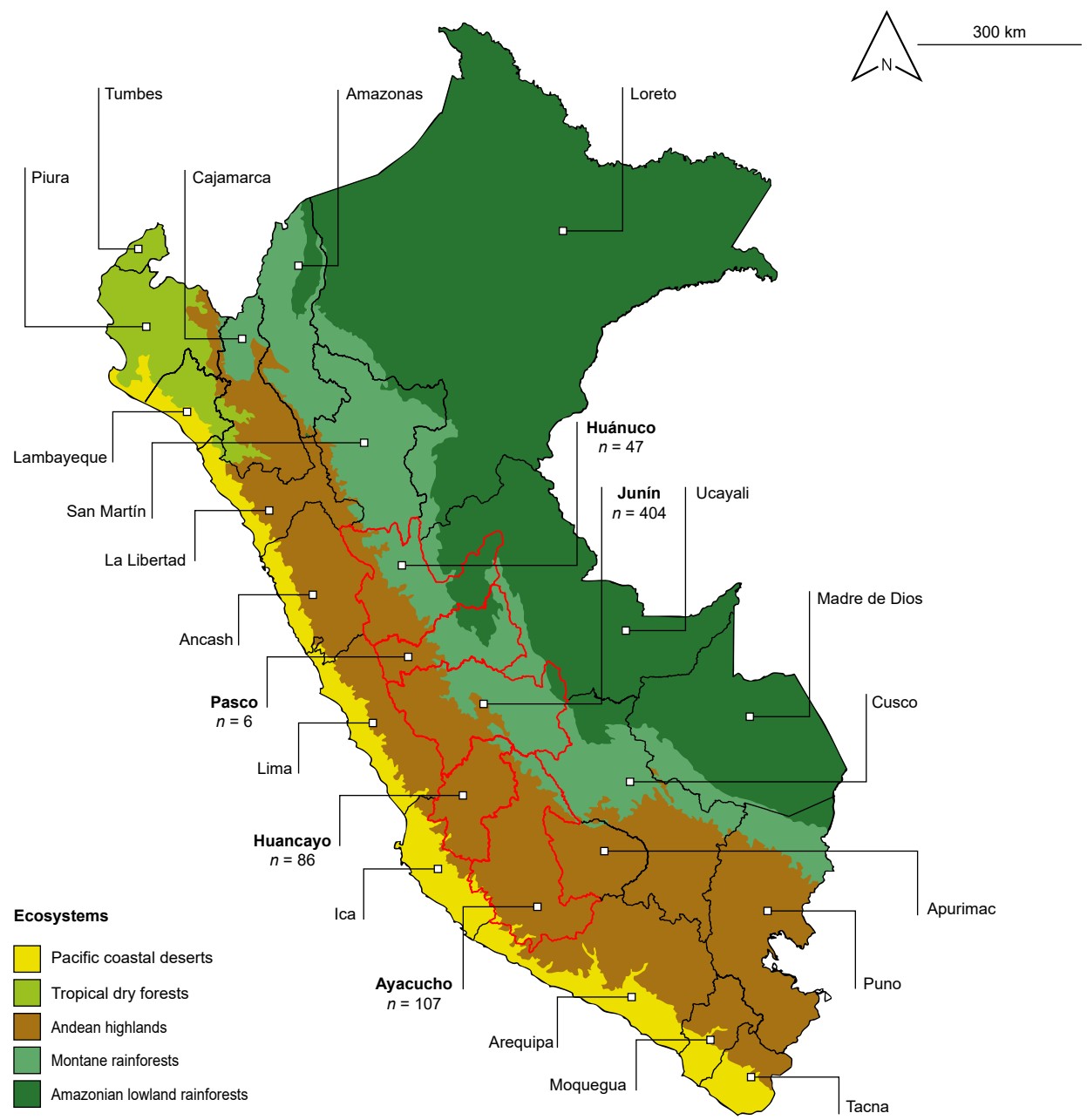

**Extended Data Fig. 1 | Andean–Amazonian regions of Peru surveyed for pesticide use.** Map of Peru showing major administrative regions. In 2020, 650 agrochemical retailers were randomly surveyed[60] across the five regions outlined in red—Ayacucho ($n = 107$), Huancayo ($n = 86$), Huánuco ($n = 47$), Junín ($n = 404$) and Pasco ($n = 6$)—selected to capture diverse agroecological conditions. For geographic context, Peru's principal ecosystems are also shown: Pacific coastal deserts (yellow), Andean highlands (brown), tropical dry forests (light green), montane rainforests (mid green) and Amazonian lowland rainforests (dark green), as classified by the Peruvian Ministry of the Environment (MINAM) in its 2019 National Ecosystem Map of Peru (https://sinia.minam.gob.pe/mapas/mapa-nacional-ecosistemas-peru; accessed 11 February 2020).

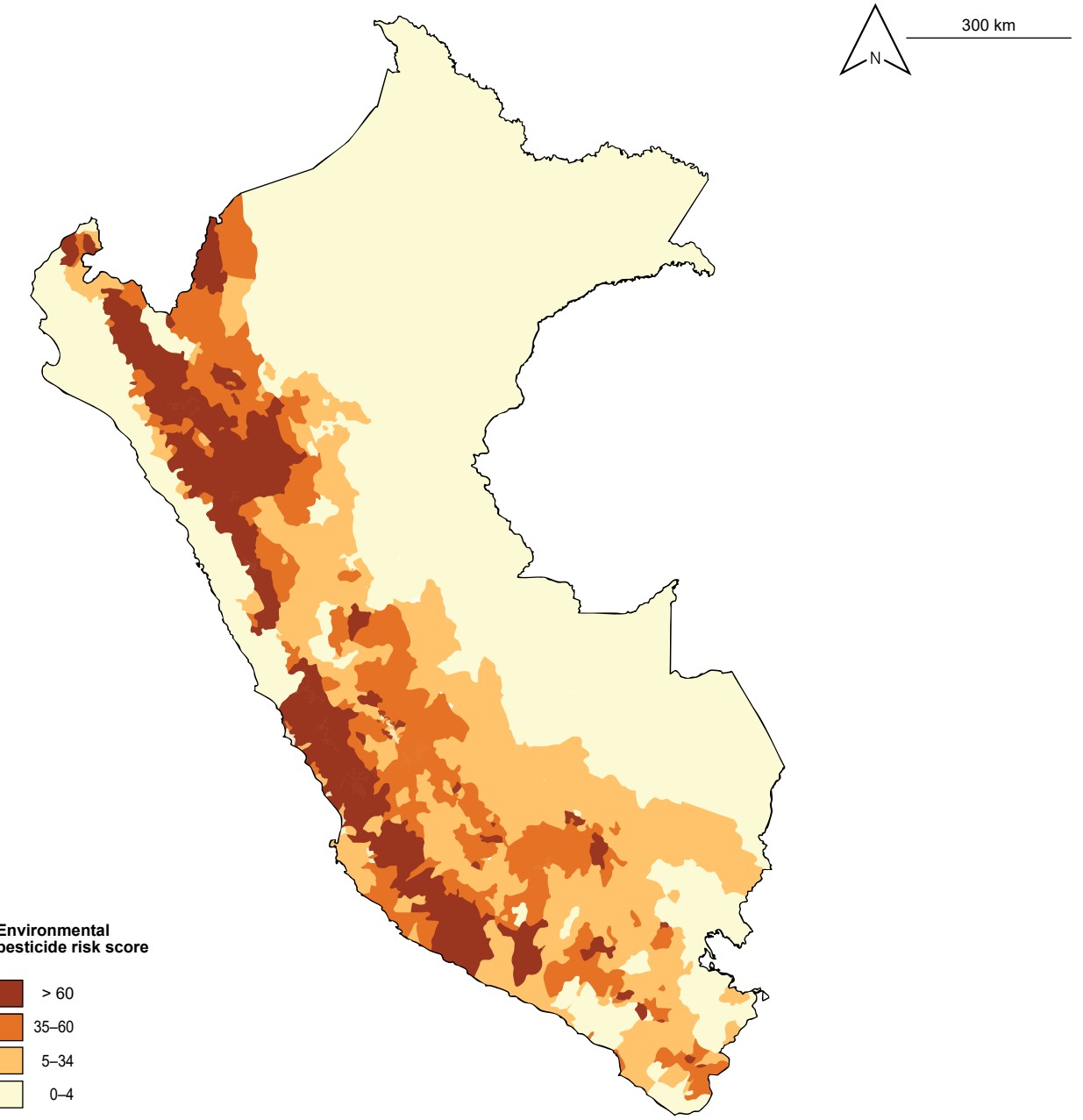

**Extended Data Fig. 2 | District-level map of modelled environmental pesticide risk in Peru, calibrated using cultivated land cover.** Choropleth map displaying pesticide risk scores (as in Fig. 1), aggregated at the district level and colour-scaled on a normalised 0–100 scale, ranging from yellow (lower risk) to brown (higher risk). Model calibration was based on the proportion of cultivated land in 2018, mapped via satellite by MINAGRI[65].

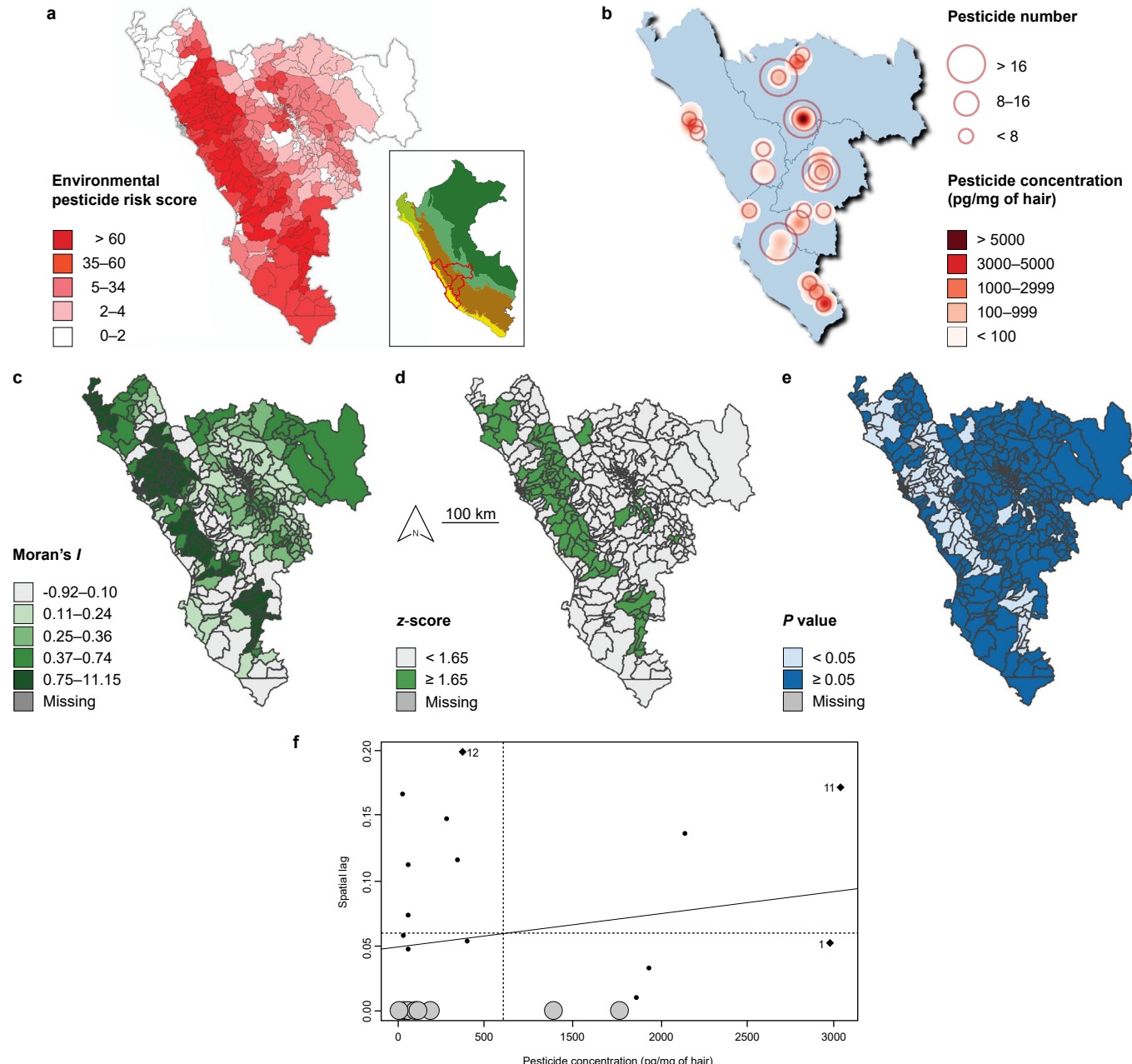

**Extended Data Fig. 3 | Spatial validation of the environmental pesticide risk model using human hair biomonitoring and spatial autocorrelation analysis.**
**a**, Choropleth map showing district-level modelled pesticide risk scores, colour-scaled in red from light (lower risk) to dark (higher risk) on a normalised 0–100 scale (as in Extended Data Fig. 2). Inset locates biomonitored regions within Peru. **b**, Proportional symbol map of the 50 sampled individuals[19]. Circle area and colour intensity (red shades) represent, respectively, the number of AIs and degradation products detected and their concentrations (pg/mg of hair) (see Methods, Supplementary Methods and Supplementary Table 1). Locations are slightly jittered to preserve privacy. **c**, Choropleth map of local bivariate Moran's *I* values (hair pesticide concentrations *vs.* modelled risk), colour-scaled in green from light to dark to indicate increasing spatial association.

**d**, Corresponding *z*-score map, colour-scaled in green from light (<1.65) to dark (≥1.65), highlighting areas of stronger spatial association. **e**, Choropleth map of associated *P* values, colour-scaled in blue from light (<0.05) to dark (≥0.05), indicating statistical significance of the spatial association. **a**–**e**, All maps refer to the same regions, located within Peru as shown on the inset in **a**. **f**, Moran scatter plot showing the relationship between measured pesticide concentrations in hair samples and their spatially lagged values from the environmental exposure risk model. The slope corresponds to global bivariate Moran's *I*, indicating positive spatial autocorrelation (Moran's *I* = 0.42; *z*-score=1.49; *P* = 0.044). Large points aligned along the *x*-axis with near-zero spatial lag identify local spatial outliers, corresponding to districts with elevated or reduced pesticide concentrations relative to neighbouring areas.

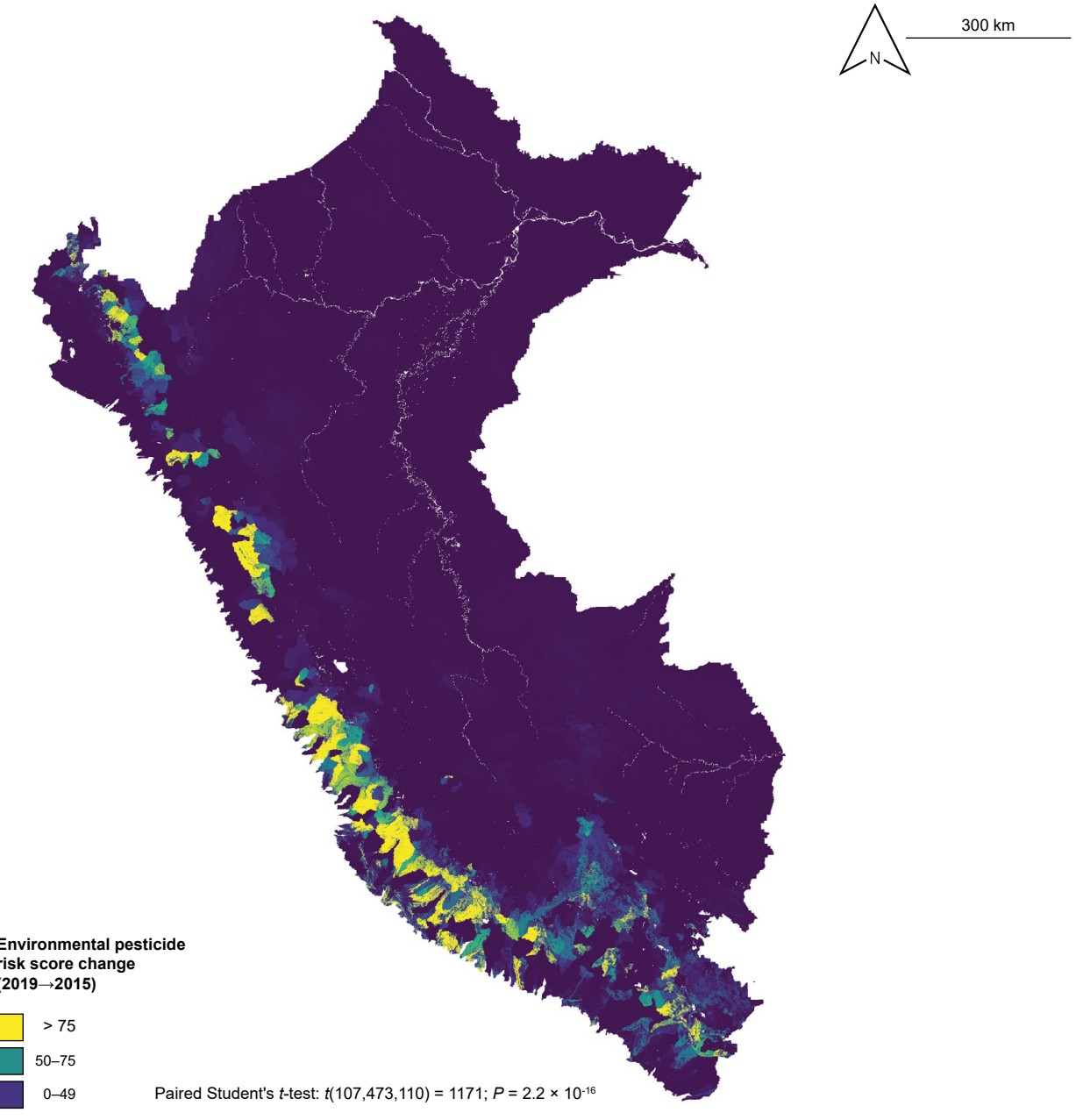

**Environmental pesticide
risk score change
(2019→2015)**

| | |
|---|---|
| 🟨 | > 75 |
| 🟩 | 50–75 |
| 🟪 | 0–49 |

Paired Student's *t*-test: $t(107,473,110) = 1171$; $P = 2.2 \times 10^{-16}$

**Extended Data Fig. 4 | Environmental pesticide exposure risk across two contrasting ENSO phases in Peru.** Raster map showing modelled changes in pesticide exposure risk between the 2015 'Godzilla' El Niño event and the 2019 neutral phase[66]. A statistically significant directional shift towards higher exposure in 2015 was detected across 124,545,177 grid cell comparisons [$t(107,473,110)=1171$; $P = 2.2 \times 10^{-16}$]. Colours indicate the magnitude of change in annual pesticide risk scores, computed on a normalised 0–100 scale (as in Fig. 1), with lighter tones representing larger differences.

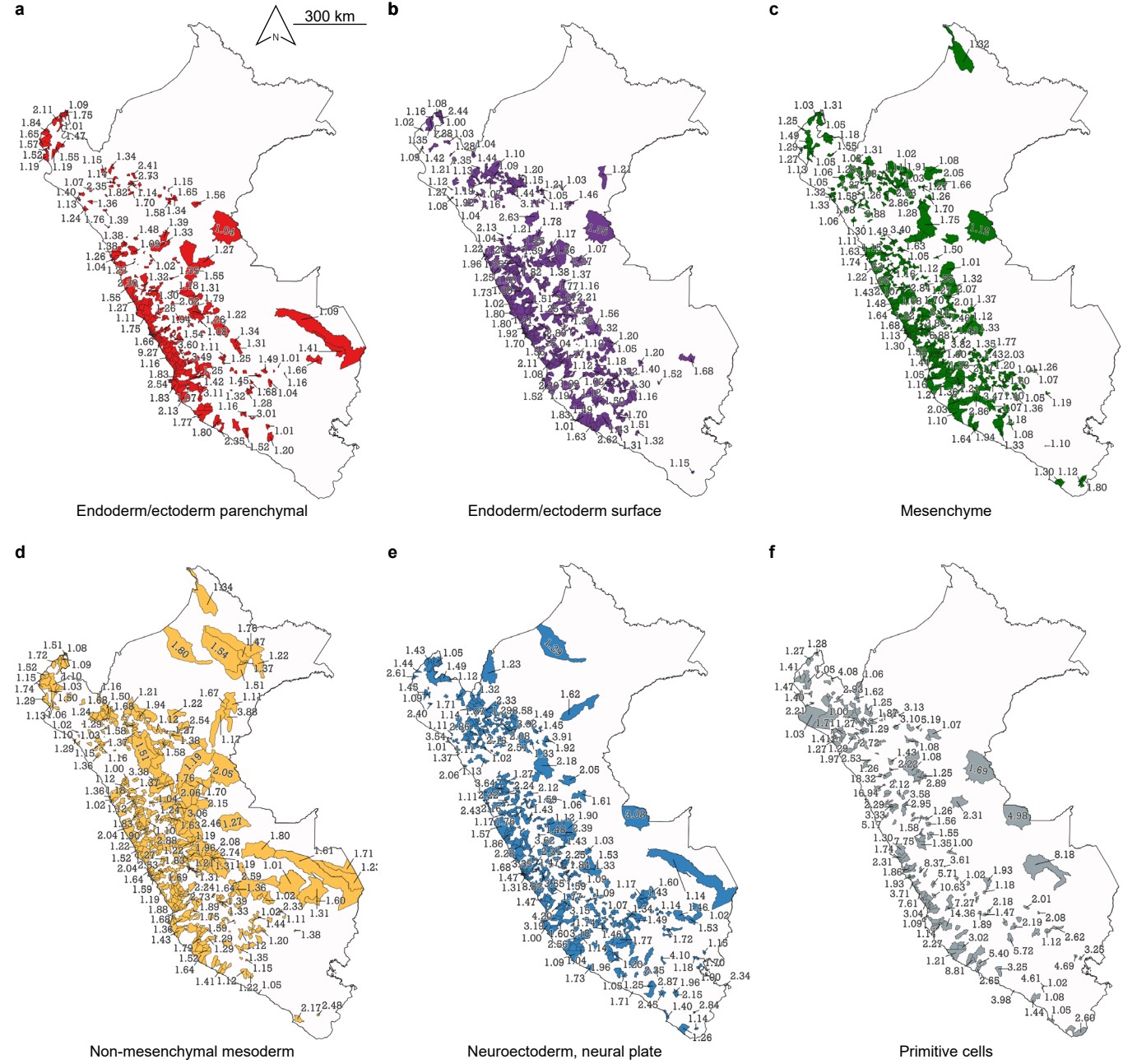

**Extended Data Fig. 5 | District-level cancer clustering in Peru based on SIRs stratified by developmental lineage.** Maps in **a–f** show spatial clusters of cancer incidence for six developmental lineages included in the geospatial modelling, based on district-level SIRs>1. SIR values are summarized visually; the complete dataset is available at https://doi.org/10.6084/m9.figshare.29728463.

**a**, endoderm/ectoderm-derived parenchyma, red (*n* = 331); **b**, endoderm/ectoderm-derived surface, violet (*n* = 541); **c**, mesenchyme, green (*n* = 514); **d**, non-mesenchymal mesoderm, yellow (*n* = 583); **e**, neuroectoderm (neural plate), blue (*n* = 557); **f**, primitive cells, grey (*n* = 224).

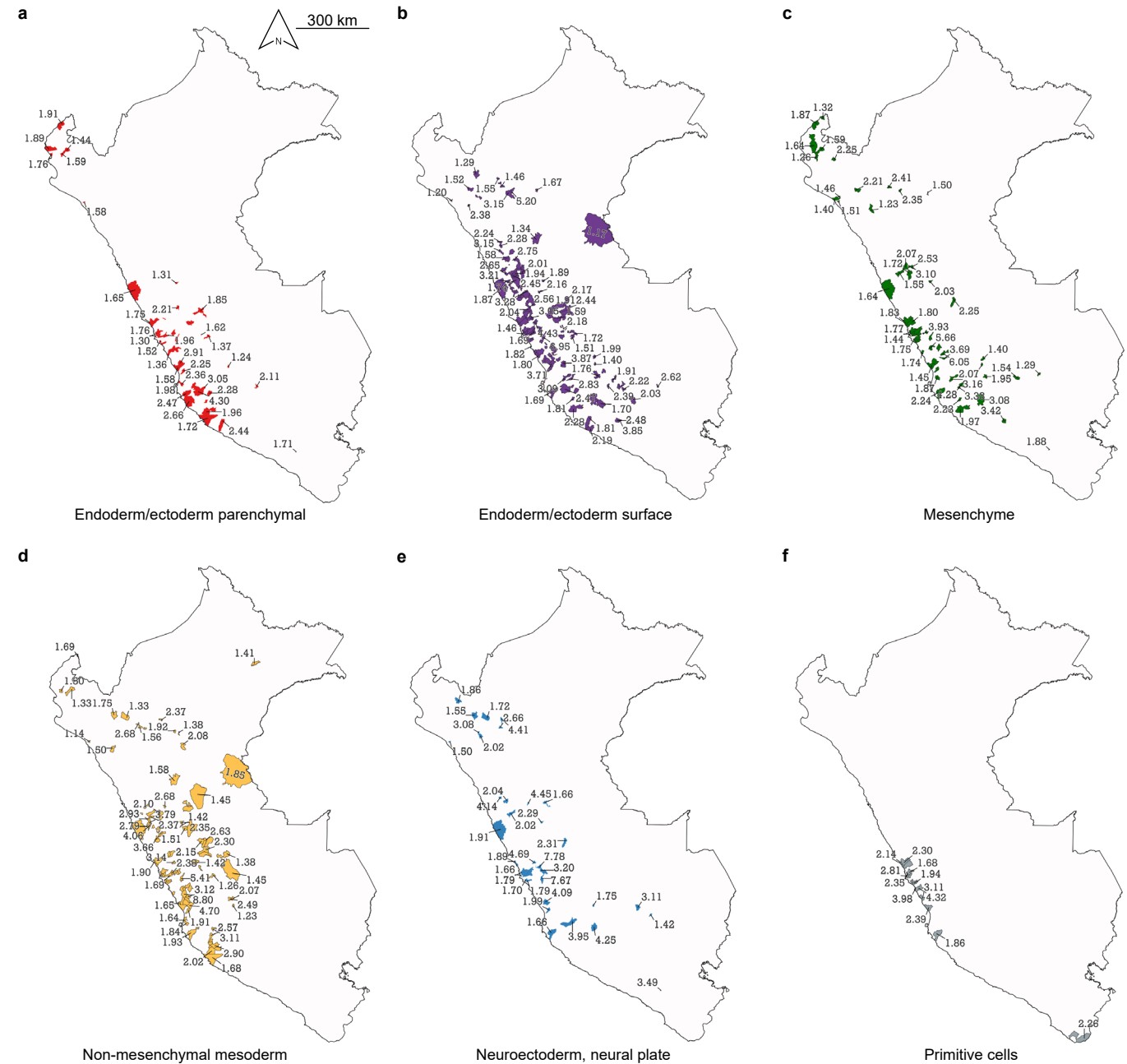

**Extended Data Fig. 6 | Pesticide-associated cancer hotspots in Peru identified through lineage-informed geostatistical modelling.** Maps in **a**–**f** show pesticide-associated cancer hotspots for six developmental lineages included in the geospatial modelling, defined as regions where the lower bound of the 95% CI for RR exceeds 1. RR values are summarized visually; the complete dataset with 95% CIs is available at https://doi.org/10.6084/m9.figshare.29728463. **a**, endoderm/ectoderm-derived parenchyma, red (*n* = 52); **b**, endoderm/ ectoderm-derived surface, violet (*n* = 156); **c**, mesenchyme, green (*n* = 64); **d**, non-mesenchymal mesoderm, yellow (*n* = 107); **e**, neuroectoderm (neural plate), blue (*n* = 40); **f**, primitive cells, grey (*n* = 17).

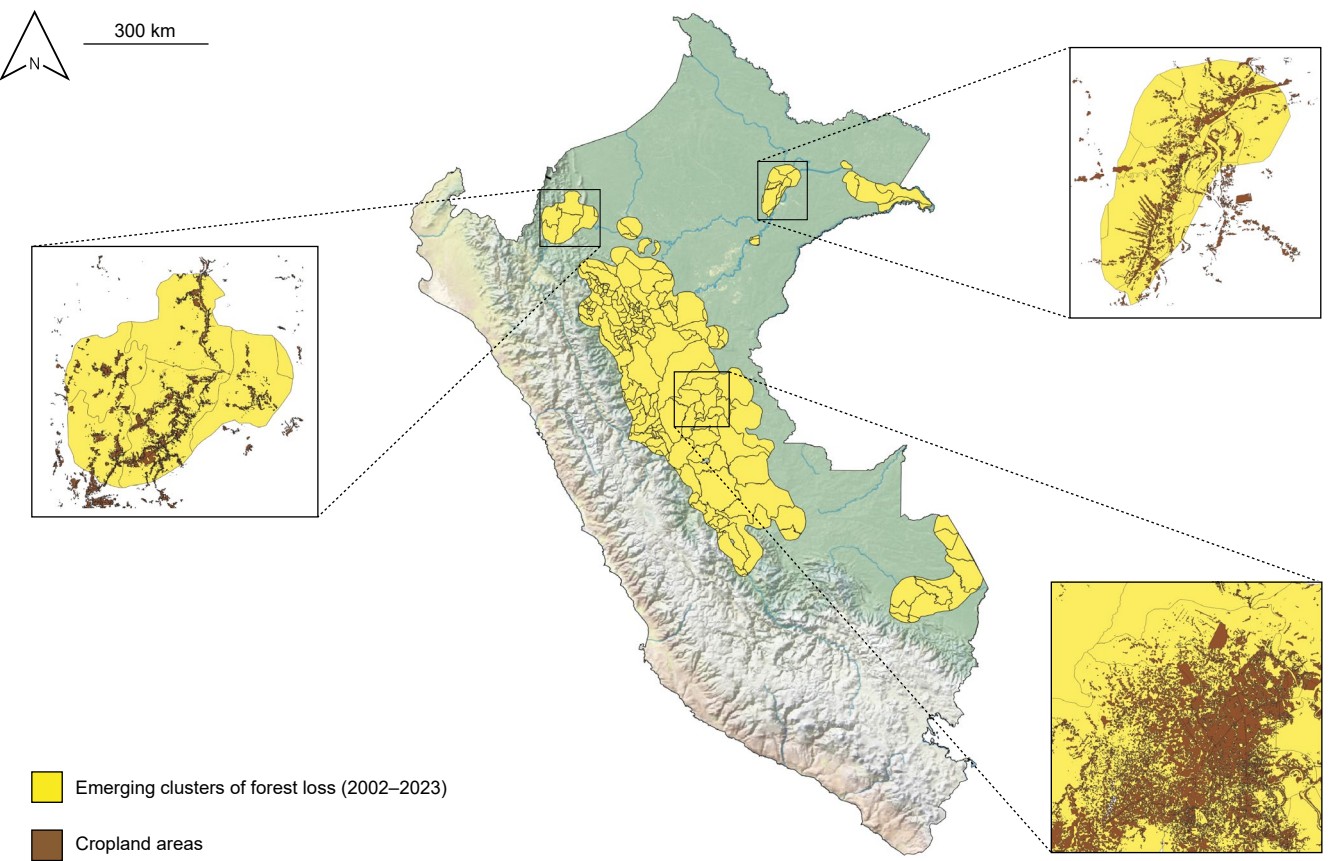

**Extended Data Fig. 7 | Spatial delineation of deforestation clusters in Peru.** The map highlights statistically significant emerging clusters of forest loss from 2002 to 2023, identified by GFW[70] (https://www.globalforestwatch.org/; accessed 23 January 2025). Yellow shading delineates cluster extent, with internal lines marking district boundaries. Incorporating these spatial data as a fixed covariate improved INLA model fit for non-mesenchymal, mesoderm-derived cancers (ΔDIC = 5.73; ΔWAIC = 3.74). Insets show selected cropland areas (brown), mapped by MINAGRI[65], within deforestation clusters attributed to recent agricultural expansion.

Legend:
- Emerging clusters of forest loss (2002–2023)
- Cropland areas

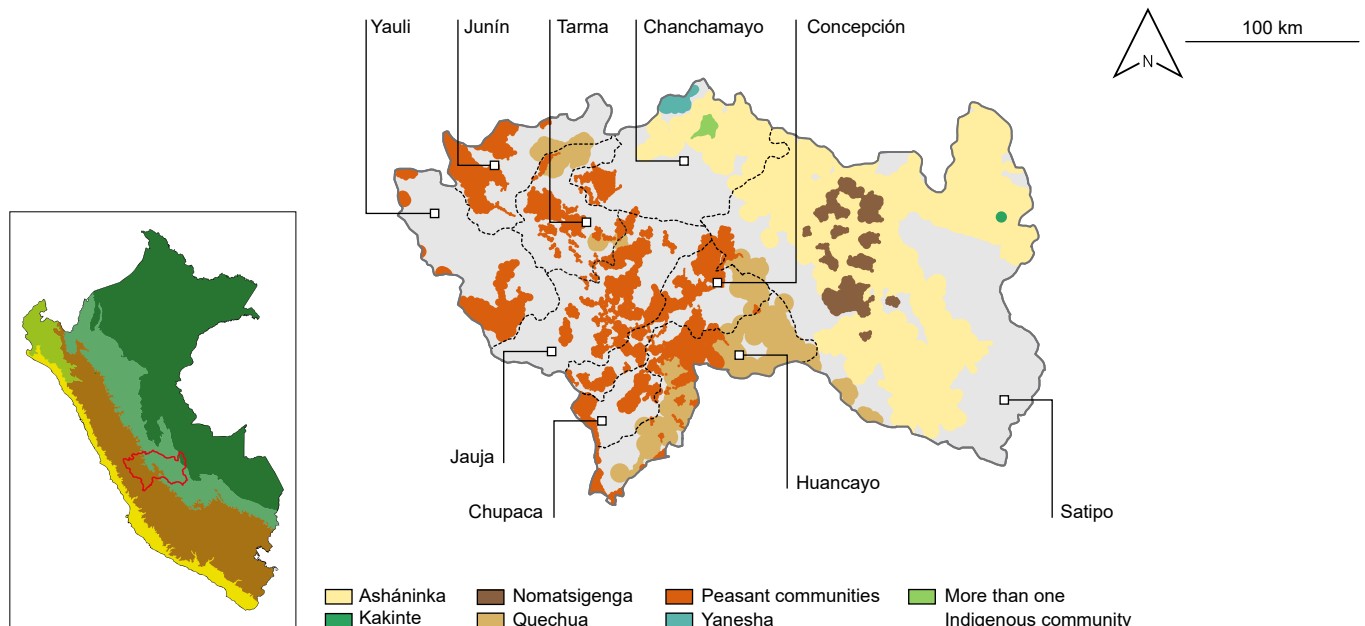

**Extended Data Fig. 8 | Territories of Indigenous and peasant communities in the Junín region.** Map showing areas inhabited by Andean–Amazonian Indigenous and peasant communities, delineated according to the identification criteria of the International Labour Organization (ILO) Convention 169 and Peruvian Law No. 29785 on the Right to Prior Consultation. Data were retrieved from the 2017 Indigenous and Peasant Communities Census[38], with the Indigenous and Native Peoples Database (BDPI; https://bdpi.cultura.gob.pe/; accessed 27 October 2019) consulted for reference. Colours indicate territories of the Asháninka (yellow), Kakinte (dark green), Nomatsigenga (brown), Quechua (beige), Yanesha (cyan), areas with more than one Indigenous community (light green) and peasant communities (orange). Provincial boundaries are shown as dotted lines. The inset locates the Junín region within Peru. Boundaries and designations are illustrative only and do not imply legal recognition of Indigenous territories or ancestral land rights.

**Extended Data Table 1 | Pesticides included in the environmental risk model, with associated physicochemical properties**

| Pesticide name | Molecular formulae | CAS registry number | PubChem Compound ID | Pesticide type | WHO hazard class | $K_{oc}$ (L/kg_OC) | DT$_{50}$Soil (days) |
|---|---|---|---|---|---|---|---|
| 2,4-Dichlorophenoxyacetic acid (2,4-D) | $C_8H_6Cl_2O_3$ | 94-75-7 | 1486 | Herbicide | II | 514.623957 | 28.8 |
| Abamectin | $C_{95}H_{142}O_{28}$ | 71751-41-2 | 6435890 | Insecticide | Ib | 45321.05385 | 30 |
| Acephate | $C_4H_{10}NO_3PS$ | 30560-19-1 | 1982 | Insecticide | II | 5501.612524 | 3 |
| Atrazine | $C_8H_{14}ClN_5$ | 1912-24-9 | 2256 | Herbicide | III | 396.551868 | 29 |
| Carbaryl | $C_{12}H_{11}NO_2$ | 63-25-2 | 6129 | Insecticide | II | 177.190323 | 16 |
| Carbofuran | $C_{12}H_{15}NO_3$ | 1563-66-2 | 2566 | Insecticide | Ib | 161.168439 | 14 |
| Chlorpyrifos (CPS) | $C_9H_{11}Cl_3NO_3PS$ | 2921-88-2 | 2730 | Insecticide | II | 83915.0775 | 27.6 |
| Copper(II) sulphate pentahydrate | $CuH_{10}O_9S$ | 7758-99-8 | 24463 | Fungicide | II | 1.873959 | 0.1 |
| Deltamethrin | $C_{22}H_{19}Br_2NO_3$ | 52918-63-5 | 40585 | Insecticide | II | 1584163.49 | 21 |
| Diazinon | $C_{12}H_{21}N_2O_3PS$ | 333-41-5 | 3017 | Insecticide | II | 5501.612524 | 18.4 |
| Difenoconazole | $C_{19}H_{17}Cl_2N_3O_3$ | 119446-68-3 | 86173 | Fungicide | II | 22263.8363 | 91.8 |
| Dimethoate | $C_5H_{12}NO_3PS_2$ | 60-51-5 | 3082 | Insecticide | II | 4.193919 | 7.2 |
| Fipronil | $C_{12}H_4Cl_2F_6N_4OS$ | 120068-37-3 | 3352 | Insecticide | II | 8629.785478 | 65 |
| Glyphosate | $C_3H_8NO_5P$ | 1071-83-6 | 3496 | Herbicide | III | 0.00021 | 6.45 |
| Imidacloprid | $C_9H_{10}ClN_5O_2$ | 138261-41-3 | 86287518 | Insecticide | II | 2.54994 | 174 |
| Linuron | $C_9H_{10}Cl_2N_2O_2$ | 330-55-2 | 9502 | Herbicide | III | 1296.582035 | 48 |
| Mancozeb | $C_{40}H_{60}Mn_9N_{20}S_{40}Zn$ | 8018-01-7 | 76957227 | Fungicide | U | 15.437252 | 0.05 |
| Metalaxyl-M | $C_{15}H_{21}NO_4$ | 70630-17-0 | 11150163 | Fungicide | II | 37.983064 | 14.1 |
| Methamidophos | $C_2H_8NO_2PS$ | 10265-92-6 | 4096 | Insecticide | Ib | 0.099266 | 4 |
| Methomyl | $C_5H_{10}N_2O_2S$ | 16752-77-5 | 4109 | Insecticide | Ib | 2.737789 | 7 |
| Metiram | $C_{16}H_{33}N_{11}S_{16}Zn_3$ | 9006-42-2 | n.a. | Fungicide | U | 1.444009 | 7 |
| Monocrotophos | $C_7H_{14}NO_5P$ | 6923-22-4 | 5371562 | Insecticide | Ib | 0.411339 | 30 |
| Novaluron | $C_{17}H_9ClF_8N_2O_4$ | 116714-46-6 | 93541 | Insecticide | U | 174916.1864 | 96.5 |
| Oxydemeton-methyl | $C_6H_{15}O_4PS_2$ | 301-12-2 | 4618 | Insecticide | Ib | 0.11443 | 5 |
| Oxyfluorfen | $C_{15}H_{11}ClF_3NO_4$ | 2874-03-3 | 39327 | Herbicide | U | 48659.76419 | 73 |
| Permethrin | $C_{21}H_{20}Cl_2O_3$ | 52645-53-1 | 40326 | Insecticide | II | 2007705.46 | 21 |
| Pirimicarb | $C_{11}H_{18}N_4O_2$ | 23103-98-2 | 31645 | Insecticide | II | 37.093687 | 9 |
| Propineb | $C_5H_8N_2S_4Zn$ | 9016-72-2 | 6100711 | Fungicide | U | 0.356829 | 4.3 |
| Sulfoxaflor | $C_{10}H_{10}F_3N_3OS$ | 946578-00-3 | 16723172 | Insecticide | II | 4.418328 | 3.54 |
| Tebuconazole | $C_{16}H_{22}ClN_3O$ | 107534-96-3 | 86102 | Fungicide | II | 4239.357093 | 47.1 |
| Trichlorfon | $C_4H_8Cl_3O_4P$ | 52-68-6 | 5853 | Insecticide | II | 2.212025 | 18 |

WHO classification of pesticides by hazard[63]: Ib, highly hazardous; II, moderately hazardous; III, slightly hazardous; U, unlikely to present an acute hazard. Abbreviations: CAS, Chemical Abstracts Service; DT$_{50}$Soil, soil dissipation half-life; $K_{oc}$, organic carbon distribution coefficient; n.a., not available; OC, organic carbon. $K_{oc}$ values are reported at full precision to preserve computational accuracy.

**Extended Data Table 2 | Gene sets for liver cancer risk factors and lineage-specific MTFs in hepatobiliary carcinogenesis**

| Symbol | Gene ID | Name | Species |
|---|---|---|---|
| **Non-genotoxic exposure signature[48]** | | | |
| ANXA2 | 302 | annexin A2 | *Homo sapiens* (human) |
| CITED4 | 163732 | Cbp/p300 interacting transactivator with Glu/Asp rich carboxy-terminal domain 4 | *Homo sapiens* (human) |
| GPR146 | 115330 | G protein-coupled receptor 146 | *Homo sapiens* (human) |
| ICA1 | 3382 | islet cell autoantigen 1 | *Homo sapiens* (human) |
| LITAF | 9516 | lipopolysaccharide induced TNF factor | *Homo sapiens* (human) |
| MAT1A | 4143 | methionine adenosyltransferase 1A | *Homo sapiens* (human) |
| PLA1A | 51365 | phospholipase A1 member A | *Homo sapiens* (human) |
| PRODH | 5625 | proline dehydrogenase 1 | *Homo sapiens* (human) |
| PSMB9 | 5698 | proteasome 20S subunit beta 9 | *Homo sapiens* (human) |
| SEL1L | 6400 | SEL1L adaptor subunit of SYVN1 ubiquitin ligase | *Homo sapiens* (human) |
| TAP1 | 6890 | transporter 1, ATP binding cassette subfamily B member | *Homo sapiens* (human) |
| TRNT1 | 51095 | tRNA nucleotidyl transferase 1 | *Homo sapiens* (human) |
| UGT2B17 | 7367 | UDP glucuronosyltransferase family 2 member B17 | *Homo sapiens* (human) |
| USP2 | 9099 | ubiquitin specific peptidase 2 | *Homo sapiens* (human) |
| **HBV infection signature[74]** | | | |
| CCL3L3 | 414062 | C-C motif chemokine ligand 3 like 3 | *Homo sapiens* (human) |
| CCL4 | 6351 | C-C motif chemokine ligand 4 | *Homo sapiens* (human) |
| CHI3L1 | 1116 | chitinase 3 like 1 | *Homo sapiens* (human) |
| CXCL10 | 3627 | C-X-C motif chemokine ligand 10 | *Homo sapiens* (human) |
| CXCL11 | 6373 | C-X-C motif chemokine ligand 11 | *Homo sapiens* (human) |
| FABP5 | 2171 | fatty acid binding protein 5 | *Homo* sapiens (human) |
| FCGR1BP | 2210 | Fc gamma receptor Ib, pseudogene | *Homo sapiens* (human) |
| GBP1 | 2633 | guanylate binding protein 1 | *Homo sapiens* (human) |
| HLA-DMA | 3108 | major histocompatibility complex, class II, DM alpha | *Homo sapiens* (human) |
| LGALS3 | 3958 | galectin 3 | *Homo sapiens* (human) |
| RNF213 | 57674 | ring finger protein 213 | *Homo sapiens* (human) |
| SLAMF7 | 57823 | SLAM family member 7 | *Homo sapiens* (human) |
| SLAMF8 | 56833 | SLAM family member 8 | *Homo sapiens* (human) |
| STAT1 | 6772 | signal transducer and activator of transcription 1 | *Homo sapiens* (human) |
| TIGAR | 57103 | TP53 induced glycolysis regulatory phosphatase | *Homo sapiens* (human) |
| **Alcohol exposure signature[75]** | | | |
| AEBP1 | 165 | AE binding protein 1 | *Homo sapiens* (human) |
| AHSG | 197 | alpha 2-HS glycoprotein | *Homo sapiens* (human) |
| ANXA1 | 301 | annexin A1 | *Homo sapiens* (human) |
| CAPN2 | 824 | calpain 2 | *Homo sapiens* (human) |
| CCL2 | 6347 | C-C motif chemokine ligand 2 | *Homo sapiens* (human) |
| CITED2 | 10370 | Cbp/p300 interacting transactivator with Glu/Asp rich carboxy-terminal domain 2 | *Homo sapiens* (human) |
| CPS1 | 1373 | carbamoyl-phosphate synthase 1 | *Homo sapiens* (human) |
| CYP3A7 | 1551 | cytochrome P450 family 3 subfamily A member 7 | *Homo sapiens* (human) |
| F2R | 2149 | coagulation factor II thrombin receptor | *Homo sapiens* (human) |
| GNB1 | 2782 | G protein subunit beta 1 | *Homo sapiens* (human) |
| ID3 | 3399 | inhibitor of DNA binding 3 | *Homo sapiens* (human) |
| LUM | 4060 | lumican | *Homo sapiens* (human) |
| MGP | 4256 | matrix Gla protein | *Homo sapiens* (human) |
| PLA2G2A | 5320 | phospholipase A2 group IIA | *Homo sapiens* (human) |
| PLIN2 | 123 | perilipin 2 | *Homo sapiens* (human) |
| RGS5 | 8490 | regulator of G protein signaling 5 | *Homo sapiens* (human) |
| S100A11 | 6282 | S100 calcium binding protein A11 | *Homo sapiens* (human) |
| TMSB10 | 9168 | thymosin beta 10 | *Homo sapiens* (human) |
| TMSB4X | 7114 | thymosin beta 4 X-linked | *Homo sapiens* (human) |
| **Steatosis signature[76]** | | | |
| ADAMTS1 | 9510 | ADAM metallopeptidase with thrombospondin type 1 motif 1 | *Homo sapiens* (human) |
| AQP1 | 358 | aquaporin 1 (Colton blood group) | *Homo sapiens* (human) |
| C1orf198 | 84886 | chromosome 1 open reading frame 198 | *Homo sapiens* (human) |
| COL1A2 | 1278 | collagen type I alpha 2 chain | *Homo sapiens* (human) |
| CRIM1 | 51232 | cysteine rich transmembrane BMP regulator 1 | *Homo sapiens* (human) |
| FRAS1 | 80144 | Fraser extracellular matrix complex subunit 1 | *Homo sapiens* (human) |
| FSTL1 | 11167 | follistatin like 1 | *Homo sapiens* (human) |
| JCAD | 57608 | junctional cadherin 5 associated | *Homo sapiens* (human) |
| MAP1B | 4131 | microtubule associated protein 1B | *Homo sapiens* (human) |
| PLXDC2 | 84898 | plexin domain containing 2 | *Homo sapiens* (human) |
| SPRY1 | 10252 | sprouty RTK signaling antagonist 1 | *Homo sapiens* (human) |
| VWF | 7450 | von Willebrand factor | *Homo sapiens* (human) |
| **Genotoxic exposure signature[48,77]** | | | |
| ADGRG1 | 9289 | adhesion G protein-coupled receptor G1 | *Homo sapiens* (human) |
| ALDH1A1 | 216 | aldehyde dehydrogenase 1 family member A1 | *Homo sapiens* (human) |
| ARL14 | 80117 | ADP ribosylation factor like GTPase 14 | *Homo sapiens* (human) |
| BTG2 | 7832 | BTG anti-proliferation factor 2 | *Homo sapiens* (human) |
| CDKN1A | 1026 | cyclin dependent kinase inhibitor 1A | *Homo sapiens* (human) |
| CES2 | 8824 | carboxylesterase 2 | *Homo sapiens* (human) |
| FOSL1 | 8061 | FOS like 1, AP-1 transcription factor subunit | *Homo sapiens* (human) |
| GDF15 | 9518 | growth differentiation factor 15 | *Homo sapiens* (human) |
| LIF | 3976 | interleukin 6 family cytokine | *Homo sapiens* (human) |
| MGMT | 4255 | O-6-methylguanine-DNA methyltransferase | *Homo sapiens* (human) |
| NCF2 | 4688 | neutrophil cytosolic factor 2 | *Homo sapiens* (human) |
| NECTIN4 | 81607 | nectin cell adhesion molecule 4 | *Homo sapiens* (human) |
| RRAD | 6236 | Ras related glycolysis inhibitor and calcium channel regulator | *Homo sapiens* (human) |
| SRC | 6714 | SRC proto-oncogene, non-receptor tyrosine kinase | *Homo sapiens* (human) |
| WNT7B | 7477 | Wnt family member 7B | *Homo sapiens* (human) |
| **Endoderm-derived MTFs (hepatobiliary lineage)[26]** | | | |
| ATF5 | 22809 | activating transcription factor 5 | *Homo sapiens* (human) |
| CEBPB | 1051 | CCAAT enhancer binding protein beta | *Homo sapiens* (human) |
| CEBPD | 1052 | CCAAT enhancer binding protein delta | *Homo sapiens* (human) |
| HNF4A | 3172 | hepatocyte nuclear factor 4 alpha | *Homo sapiens* (human) |
| ID2 | 3398 | inhibitor of DNA binding 2 | *Homo sapiens* (human) |
| KLF9 | 687 | KLF transcription factor 9 | *Homo sapiens* (human) |
| MLXIPL | 51085 | MLX interacting protein like | *Homo sapiens* (human) |
| ONECUT2 | 9480 | one cut homeobox 2 | *Homo sapiens* (human) |
| ZGPAT | 84619 | zinc finger CCCH-type and G-patch domain containing | *Homo sapiens* (human) |

# Reporting Summary

## Statistics

For all statistical analyses, confirm that the following items are present in the figure legend, table legend, main text, or Methods section.

| n/a | Confirmed | |
|---|---|---|
| ☐ | ☒ | The exact sample size ($n$) for each experimental group/condition, given as a discrete number and unit of measurement |
| ☒ | ☐ | A statement on whether measurements were taken from distinct samples or whether the same sample was measured repeatedly |
| ☐ | ☒ | The statistical test(s) used AND whether they are one- or two-sided<br>*Only common tests should be described solely by name; describe more complex techniques in the Methods section.* |
| ☐ | ☒ | A description of all covariates tested |
| ☐ | ☒ | A description of any assumptions or corrections, such as tests of normality and adjustment for multiple comparisons |
| ☐ | ☒ | A full description of the statistical parameters including central tendency (e.g. means) or other basic estimates (e.g. regression coefficient) AND variation (e.g. standard deviation) or associated estimates of uncertainty (e.g. confidence intervals) |
| ☐ | ☒ | For null hypothesis testing, the test statistic (e.g. $F$, $t$, $r$) with confidence intervals, effect sizes, degrees of freedom and $P$ value noted<br>*Give P values as exact values whenever suitable.* |
| ☐ | ☒ | For Bayesian analysis, information on the choice of priors and Markov chain Monte Carlo settings |
| ☐ | ☒ | For hierarchical and complex designs, identification of the appropriate level for tests and full reporting of outcomes |
| ☒ | ☐ | Estimates of effect sizes (e.g. Cohen's $d$, Pearson's $r$), indicating how they were calculated |

*Our web collection on statistics for biologists contains articles on many of the points above.*

## Software and code

Policy information about availability of computer code

| Data collection | All open-source datasets used in this study, including version details and access information, are provided in Supplementary Table 2. Additionally, KoboToolbox (v2.020.25) and OpenRefine (v3.4.1) were used for data collection and data cleaning, respectively. |
|---|---|
| Data analysis | The custom R scripts for generating the process-based pesticide exposure risk map and for integrated nested Laplace approximation (INLA) modelling are available in the Figshare repository at https://doi.org/10.6084/m9.figshare.29728463. Access credentials for a test version of the code are available from the corresponding author upon reasonable request. All other analyses relied on available software, fully referenced in the manuscript, including:<br>- Bioconductor (v.3.20),<br>- Excel (v.16.16.27),<br>- Fred's Softwares,<br>- GraphPad Prism (v.10.4.1),<br>- Mitomaster (v.Beta 1),<br>- Nominatim (v.4.5.0),<br>- PostgreSQL (v.16.3),<br>- Python (v.3.11.3),<br>- QGIS (v.3.28 LTR), and<br>- QuantaSoft (v.1.7)<br>- R (v.4.4.1), R packages included: dplyr (v.1.1.4), gghalves (v.0.1.4), ggplot2 (v.3.5.2), ggrain (v.1.0.2), INLA (v.25.3.24), Plotly (v.4.10.4), raincloudplots (v.0.2.0), sf (v.1.0-10), sp (v.1.5-1), SpatialEpi (v.1.2.8), spdep (v.1.3-10), sva (v.3.54.0), terra (v.1.8-70), and tmap (v.4.0). |

For manuscripts utilizing custom algorithms or software that are central to the research but not yet described in published literature, software must be made available to editors and reviewers. We strongly encourage code deposition in a community repository (e.g. GitHub). See the Nature Portfolio guidelines for submitting code & software for further information.

## Data

Policy information about availability of data

All manuscripts must include a data availability statement. This statement should provide the following information, where applicable:

- Accession codes, unique identifiers, or web links for publicly available datasets
- A description of any restrictions on data availability
- For clinical datasets or third party data, please ensure that the statement adheres to our policy

> The datasets generated and analysed during the current study are available in the Figshare repository at https://doi.org/10.6084/m9.figshare.29728463, except for individual-level geolocation data from GEO and the National Cancer Institute of Peru (INEN) Cancer Registry, which are not publicly available to protect privacy and comply with data protection regulations. Transcriptomic data analysed are publicly available in GEO under accession numbers GSE111580 and GSE136247 (Peru). Comparative datasets were obtained from GEO under accession numbers GSE17548 (Turkey), GSE45436 (Taiwan), and GSE62232 (France). No new individual-level data were collected, and informed consent was not required.

## Research involving human participants, their data, or biological material

Policy information about studies with human participants or human data. See also policy information about sex, gender (identity/presentation), and sexual orientation and race, ethnicity and racism.

| | |
|---|---|
| Reporting on sex and gender | Sex was recorded from medical records and corresponds to biological sex assigned at diagnosis; gender identity was not documented. Although the primary objective focused on spatial patterns of tumour incidence and environmental exposures irrespective of sex or gender, stratification by sex (female, male) and age group (0–39, 40–59, ≥60 years) was applied when calculating standardized incidence ratios (SIRs). No further sex- or gender-based stratification was conducted in the geospatial or transcriptomic analyses, and all findings apply to both sexes. Sex-disaggregated clinical epidemiological data for the Peruvian liver cancer cohort are available; individual-level data are included only where participant consent and ethics approval permitted data sharing. |
| Reporting on race, ethnicity, or other socially relevant groupings | Ethnicity, race, or other socially defined categories were not used as variables in the study design or analyses. Such groupings were not incorporated in geospatial or transcriptomic analyses, and no adjustments for them were performed. However, in the Peruvian liver cancer cohort, matrilineal ancestry data were included to provide context on genetic background and population structure. These were inferred by researchers through mitochondrial haplogrouping, based on the established classification of Native American haplotypes (A–D) from reference sequences. These data were not interpreted as proxies for social constructs. |
| Population characteristics | Covariate-relevant population characteristics included age, sex, residential address at diagnosis, and primary cancer diagnosis (ICD-10 codes C00–C96), obtained from the INEN cancer registry. These variables informed stratification of cancer incidence ratios and geospatial modelling. Additionally, clinical records of 36 Peruvian liver cancer patients were reviewed to extract covariates relevant to aetiology and progression, including hepatitis B status, history of alcohol use or metabolic disorders (e.g. steatosis or diabetes), and any treatment administered. |
| Recruitment | Participants were not actively recruited; rather, the study retrospectively included all eligible cases recorded in the INEN cancer registry from 2007 to 2020. INEN holds the most comprehensive oncology dataset in the country, comprising cases diagnosed and treated at the national referral centre. Potential sources of bias include referral and diagnostic imbalances due to the under-representation of remote or underserved regions with limited access to specialist care, and possible under-ascertainment of early-stage or non-histologically confirmed cancers. |
| Ethics oversight | The Cancer Registry at INEN is authorised by the National Authority for the Protection of Personal Data (ANPD) under registration number RNPDP-EP 4794. The study also received approval from the INEN Institutional Review Board under reference numbers 113-2014-CIE/INEN, 407-2016-CIE/INEN, and 049-2024-DICON/INEN. |

Note that full information on the approval of the study protocol must also be provided in the manuscript.

# Field-specific reporting

Please select the one below that is the best fit for your research. If you are not sure, read the appropriate sections before making your selection.

☒ Life sciences          ☐ Behavioural & social sciences          ☐ Ecological, evolutionary & environmental sciences

For a reference copy of the document with all sections, see nature.com/documents/nr-reporting-summary-flat.pdf

# Life sciences study design

All studies must disclose on these points even when the disclosure is negative.

| | |
|---|---|
| Sample size | No sample-size calculation was performed. Instead, we included all eligible cancer cases recorded in the INEN registry from 2007 to 2020. This period was selected to capture trends consistent with chronic environmental exposures, including pesticides, and to align broadly with the modelled pesticide risk window (2014–2019). After applying exclusion criteria to ensure data quality, the final dataset comprised 158,072 primary cancer cases (C00–C96), offering robust statistical power for spatial and lineage-specific analyses. |

| Data exclusions | Exclusion criteria were pre-established to ensure data integrity and geographic precision. Recurrent cases, duplicates, and internally inconsistent records were removed. To minimise misclassification of chronic exposure, individuals without verified residency at the recorded address for at least five years prior to diagnosis were excluded. These criteria ensured the inclusion of primary cancer cases with robust diagnostic and spatial data suitable for geostatistical analyses. |
|---|---|
| Replication | This study relies on retrospective analysis of existing registry and environmental data; no experimental procedures were performed that require replication. All inclusion criteria and analytical methods are fully documented to enable reproducibility using comparable datasets. |
| Randomization | This study did not involve random allocation, as all cancer cases were retrospectively analysed. Tumours were assigned post hoc to biologically coherent groups based on the developmental lineage classification of neoplasms introduced by Jules J. Berman. To reconcile biological relevance with statistical robustness, tumours were stratified at the fifth hierarchical level of this taxonomy. Spatial models accounted for potential confounding through structured and unstructured random effects. |
| Blinding | Blinding was not relevant to this study, as no experimental interventions or subjective outcome assessments were involved. |

# Reporting for specific materials, systems and methods

We require information from authors about some types of materials, experimental systems and methods used in many studies. Here, indicate whether each material, system or method listed is relevant to your study. If you are not sure if a list item applies to your research, read the appropriate section before selecting a response.

## Materials & experimental systems

| n/a | Involved in the study |
|---|---|
| ☒ | ☐ Antibodies |
| ☒ | ☐ Eukaryotic cell lines |
| ☒ | ☐ Palaeontology and archaeology |
| ☒ | ☐ Animals and other organisms |
| ☐ | ☒ Clinical data |
| ☒ | ☐ Dual use research of concern |
| ☒ | ☐ Plants |

## Methods

| n/a | Involved in the study |
|---|---|
| ☒ | ☐ ChIP-seq |
| ☒ | ☐ Flow cytometry |
| ☒ | ☐ MRI-based neuroimaging |

## Clinical data

Policy information about clinical studies

All manuscripts should comply with the ICMJE guidelines for publication of clinical research and a completed CONSORT checklist must be included with all submissions.

| Clinical trial registration | Not applicable. |
|---|---|
| Study protocol | Not applicable. |
| Data collection | Data were collected from the cancer registry maintained by INEN in Lima (Peru), covering the period 2007–2020. |
| Outcomes | Not applicable. |

## Plants

| Seed stocks | *Report on the source of all seed stocks or other plant material used. If applicable, state the seed stock centre and catalogue number. If plant specimens were collected from the field, describe the collection location, date and sampling procedures.* |
|---|---|
| Novel plant genotypes | *Describe the methods by which all novel plant genotypes were produced. This includes those generated by transgenic approaches, gene editing, chemical/radiation-based mutagenesis and hybridization. For transgenic lines, describe the transformation method, the number of independent lines analyzed and the generation upon which experiments were performed. For gene-edited lines, describe the editor used, the endogenous sequence targeted for editing, the targeting guide RNA sequence (if applicable) and how the editor was applied.* |
| Authentication | *Describe any authentication procedures for each seed stock used or novel genotype generated. Describe any experiments used to assess the effect of a mutation and, where applicable, how potential secondary effects (e.g. second site T-DNA insertions, mosiacism, off-target gene editing) were examined.* |

