## [Peer Review File · Nature Health]

Mapping Pesticide Mixtures to Cancer Risk at Country Scale with Spatial Exposomics

Corresponding Author: Dr Stéphane Bertani

Version 0:

Reviewer comments:

Reviewer #1

(Remarks to the Author)

The authors used a comprehensive spatial Bayesian framework to combine high-resolution modeling of pesticide risks with extensive cancer registry data. This allowed them to map out areas in Peru where cancer is linked to pesticide use with unprecedented accuracy. Their model looks at 31 key active ingredients in pesticides, and through innovative stratification of cancer case development, they revealed a strong connection between exposure to environmental pesticides and cancer rates.

The data that has been meticulously collected from a diverse range of sources is of good quality. Multiple verification steps have been implemented to ensure its accuracy and integrity. The sampling methods employed were carefully designed to be representative of the entire population under study, covering various demographics and relevant factors.

Regarding the analysis methods, they are appropriate for the nature of the data. A comprehensive set of statistical techniques and algorithms have been utilized, taking into account the specific characteristics and requirements of the research question. These methods have been thoroughly tested and validated against established standards and benchmarks in the field.

The presentation of the findings is attractive. The visual aids, such as graphs, charts, and diagrams, have been carefully crafted to clearly convey the key information. The layout is well-organized, with a logical flow that guides the reader through the data and analysis. The use of colors, fonts, and labels is consistent and easy to read, enhancing the overall clarity and readability of the presentation.

The conclusions drawn from the analysis are reliable. They are firmly based on the data and the results of the analysis. Multiple lines of evidence have been considered, and potential sources of error or bias have been carefully addressed. The conclusions have also been cross-checked and compared with existing research in the area to ensure their consistency and validity.

(Remarks on code availability)

After installing the required packages "INLA", "ggplot2", "dplyr", "SpatialEpi", "spdep", "tmap", "terra", we ran the code "INLA_model.R" in RStudio. However, the file "DistritosPeruRiskAvNorm.shp" required to run the code "INLA_model.R" is not available in the provided data.

The packages "sp", "sf", "rgdal", "raster", "terra" are required to run the code "Pesticide_exposure_avg.R". However, the package "rgdal" is not available for our current version of R. Also, the file "KockKd.csv" required to run the code "Pesticide_exposure_avg.R" is not available in the provided data.

It is recommended that the code and related data are better organized to help reproduce the results.

(Remarks on figshare data availability)

The cancer dataset, the environmental pesticide exposure risk map, and the Standardised Incidence Ratios (SIRs) and Relative Risk (RR) at the district level are available to help reproduce the results and support the conclusions.

Reviewer #2

(Remarks to the Author)

The manuscript describes an ambitious attempt to use spatial exposomics to identify mixtures of pesticides that cause cancer in humans. To accomplish its goal, agrochemical retailers in various regions of Peru were surveyed about pesticide names and application frequencies. In total 650 retailers were interviewed. These data were used to model exposure from 2014-2019. A pesticide exposure model was developed to estimate the spatial concentrations of 19 active ingredients. This model incorporated a number of factors that influence environmental transport of pesticides, including geography, climate factors, runoff-to-precipitation ratios, and soil dissipation half-life. Cancer registry data from 2007 to 2020 were used to ascertain cancer occurrence in Peru and to classify cancers by developmental lineage. In 2020, 50 adults were selected to donate a hair sample that was used to assay pesticide concentrations. In addition, 39 surgical specimens (tumor and paired normal adjacent tissue) from hepatocellular carcinoma patients were collected for transcriptomic profiling. The authors claim to have found compelling evidence of a mechanistic link between pesticide exposures and cancer. They further conclude that this discovery and approach is paradigm shifting that redefines the exposomic frameworks and offers a holistic understanding of how complex pesticide mixtures potentiate carcinogenesis.

Comments:

- 1) This is an ecological study prone to ecologic bias. As such, we do not know if the people diagnosed with a cancer were actually exposed. The potential for ecologic bias should be investigated and at least discussed in the discussion.
- 2) The exposure assessment is inadequate. Pesticide exposure estimated from retailers' recall of sales of specific pesticides for which only frequency of application was asked. To comprehensively reconstruct exposure duration and intensity, this study only has frequency of application, but the amount of pesticide used (intensity) and duration of each use is unknown.
- 3) Exposure was reconstructed for a six-year time frame (2014-2019). However, cancer cases that were diagnosed prior to this exposure were included in the analysis. The cancers that occurred in that seven-year period (2007-2013) cannot be caused by the exposure that were reconstructed for 2014 and should not be included in the analyses. Furthermore, cancers of the lung, breast, colon and rectum, prostate, have a long empirical latent period between exposure and diagnosis. As such, a six-year period of exposure reconstruction is likely inadequate to capture the biologically relevant exposure window.
- 4) The grouping of cancer sites by developmental lineage is an interesting hypothesis. However, I am unaware of evidence that such groupings are etiologically relevant. If there is such literature, citing it would be useful to readers. On the other hand, there are compelling examples that causes (and risk factors) are not uniform by developmental lineage. For instance, premenopausal and postmenopausal breast cancers arise from the same developmental lineage, but have unique risk factor profiles (e.g., obesity).
- 5) The standardized ratio mapping only considers biological sex and age as confounding factors, raising the possibility and likelihood that there is uncontrolled confounding for tobacco and alcohol use and other lifestyle factors.
- 6) The transcriptomic profiling of 36 hepatocellular carcinoma patients is small, and it is unclear to what extent they were actually exposed to a mixture of pesticides, rendering the results intriguing, but not compelling.

(Remarks on code availability)

(Remarks on figshare data availability)

Reviewer #3

(Remarks to the Author)

The authors conducted a spatial study to identify the association between pesticide exposure and cancer hotspots in Peru. To achieve this aim, the authors used Bayesian approaches to determine spatial clustering of cancer through the estimation of standardized incidence ratios (SIRs). They further divided the neoplasms based on their histological origin and found several clusters associated with pesticide exposure. This is an important study and a novel contribution to the literature of cancer etiology and exposures. Peru is a South American country with substantial agricultural activity. Identifying links between cancer development and pesticide exposure is of major public health relevance. I agree with the approach and conclusions. These data could explain the high incidence of blood cancers in northern Peru, which has been historically reported.

Major revisions

- Please add a limitation paragraph
- I would be cautious in the following assertive sentence: "Our findings provide compelling evidence that pesticide exposure is a principal driver of liver cancer in this population." (Discussion section, Page 18, Lines 405-406). I would reduce the assertive language. Although the authors identified unique signatures in the Peruvian samples compared to the French samples, the study design does not allow for this conclusion. Additionally, alcohol consumption in Peru is likely underestimated, and residual confounding is always a possibility.
- Comment on the link between pesticides and blood cancer etiology. Previous studies in the US have identified that living near industrial factories is linked with a higher incidence of leukemia among children. Could these pesticides also be associated with the clusters of heme malignancies identified in the study?

(Remarks on code availability)

(Remarks on figshare data availability)

Version 1:

Reviewer comments:

Reviewer #2

(Remarks to the Author)

(Remarks on code availability)

(Remarks on figshare data availability)

Spatial exposomics reveals pesticide mixtures as drivers of cancer risk

Point-by-point Response to Referees

Response Guide

Reviewer #1 (Remarks to the Author)

Response to Reviewer #1

Reviewer #2 (Remarks to the Author)

Response to Reviewer #2

Reviewer #3 (Remarks to the Author)

Response to Reviewer #3

Rebuttal Fig. 1

Rebuttal Fig. 2

Rebuttal Fig. 3

Rebuttal Fig. 4

Rebuttal Fig. 5

Rebuttal References

Reviewer #1 (Remarks to the Author):

The authors used a comprehensive spatial Bayesian framework to combine high-resolution modeling of pesticide risks with extensive cancer registry data. This allowed them to map out areas in Peru where cancer is linked to pesticide use with unprecedented accuracy. Their model looks at 31 key active ingredients in pesticides, and through innovative stratification of cancer case development, they revealed a strong connection between exposure to environmental pesticides and cancer rates.

The data that has been meticulously collected from a diverse range of sources is of good quality. Multiple verification steps have been implemented to ensure its accuracy and integrity. The sampling methods employed were carefully designed to be representative of the entire population under study, covering various demographics and relevant factors.

Regarding the analysis methods, they are appropriate for the nature of the data. A comprehensive set of statistical techniques and algorithms have been utilized, taking into account the specific characteristics and requirements of the research question. These methods have been thoroughly tested and validated against established standards and benchmarks in the field.

The presentation of the findings is attractive. The visual aids, such as graphs, charts, and diagrams, have been carefully crafted to clearly convey the key information. The layout is well-organized, with a logical flow that guides the reader through the data and analysis. The use of colors, fonts, and labels is consistent and easy to read, enhancing the overall clarity and readability of the presentation.

The conclusions drawn from the analysis are reliable. They are firmly based on the data and the results of the analysis. Multiple lines of evidence have been considered, and potential sources of error or bias have been carefully addressed. The conclusions have also been cross-checked and compared with existing research in the area to ensure their consistency and validity.

Reviewer #1 (Remarks on code availability):

After installing the required packages “INLA”, “ggplot2”, “dplyr”, “SpatialEpi”, “spdep”, “tmap”, “terra”, we ran the code “INLA_model.R” in RStudio. However, the file “DistritosPeruRiskAvNorm.shp” required to run the code “INLA_model.R” is not available in the provided data.

The packages “sp”, “sf”, “rgdal”, “raster”, “terra” are required to run the code “Pesticide_exposure_avg.R”. However, the package “rgdal” is not available for our current version of R. Also, the file “KocKd.csv” required to run the code “Pesticide_exposure_avg.R” is not available in the provided data.

It is recommended that the code and related data are better organized to help reproduce the results.

Reviewer #1 (Remarks on figshare data availability):

The cancer dataset, the environmental pesticide exposure risk map, and the Standardised Incidence Ratios (SIRs) and Relative Risk (RR) at the district level are available to help reproduce the results and support the conclusions.

Response to Reviewer #1

We thank Reviewer #1 for the careful and thorough evaluation of our study and supporting materials. We appreciate the recognition of the methodological rigour, analytical soundness, and clarity of presentation that underpin our work—elements we consider fundamental to the robustness, consistency, and credibility of our conclusions. We fully concur with Reviewer #1 on the importance of transparency and reproducibility in environmental health research and

have reorganised the code and associated datasets (Supplementary .zip archive), complemented by an interactive *Shiny* application that offers stepwise exploration of the workflow and enables reproducible execution of the analyses (accessible via: <http://doi.org/10.6084/m9.figshare.30319228>, with reviewer credentials provided to the Editor in the cover letter).

Comment #1.1: After installing the required packages “INLA”, “ggplot2”, “dplyr”, “SpatialEpi”, “spdep”, “tmap”, “terra”, we ran the code “INLA_model.R” in RStudio. However, the file “DistritosPeruRiskAvNorm.shp” required to run the code “INLA_model.R” is not available in the provided data.

In accordance with Reviewer #1’s observation, the file “DistritosPeruRiskAvNorm.shp” required to run “INLA_model.R” has now been added to the Figshare data deposit, accessible at: <https://doi.org/10.6084/m9.figshare.29728463> (or via the associated private link).

Comment #1.2: The packages “sp”, “sf”, “rgdal”, “raster”, “terra” are required to run the code “Pesticide_exposure_avg.R”. However, the package “rgdal” is not available for our current version of R.

As noted by Reviewer #1, the “rgdal” package is no longer maintained but was fully functional during the period of our modelling. As an alternative, the “terra” package (v.1.8-70) performs equivalent geospatial operations and has been implemented in a *Shiny* application (Pesticide Risk tab) included with this revision, accessible via: <http://doi.org/10.6084/m9.figshare.30319228>.

Comment #1.3: Also, the file “KocKd.csv” required to run the code “Pesticide_exposure_avg.R” is not available in the provided data.

In accordance with Reviewer #1’s observation, the file “KocKd.csv” required to run “Pesticide_exposure_avg.R” has now been added to the Figshare data deposit, available at: <https://doi.org/10.6084/m9.figshare.29728463> (or via the associated private link). To ensure compliance with data protection regulations and safeguard participant privacy, all potentially sensitive or personally identifiable information has been masked or excluded.

Comment #1.4: It is recommended that the code and related data are better organized to help reproduce the results.

As outlined in our preamble to Reviewer #1, we have substantially improved the organisation of the code and associated datasets (Supplementary .zip archive). We implemented an interactive *Shiny* application comprising two modules (*i.e.*, INLA model and Pesticide Risk) that provide transparent, stepwise visualisation of the analytical workflow and fully reproducible execution of the core analyses. For demonstration, we include the mesoderm lineage and a representative insecticide (*i.e.*, methomyl), modelled over the three-month period from January to March 2014, as exemplars; accessible via: <http://doi.org/10.6084/m9.figshare.30319228>.

Reviewer #2 (Remarks to the Author):

The manuscript describes an ambitious attempt to use spatial exposomics to identify mixtures of pesticides that cause cancer in humans. To accomplish its goal, agrochemical retailers in various regions of Peru were surveyed about pesticide names and application frequencies. In total 650 retailers were interviewed. These data were used to model exposure from 2014-2019. A pesticide exposure model was developed to estimate the spatial concentrations of 19 active ingredients. This model incorporated a number of factors that influence environmental transport of pesticides, including geography, climate factors, runoff-to-precipitation ratios, and soil dissipation half-life. Cancer registry data from 2007 to 2020 were used to ascertain cancer occurrence in Peru and to classify cancers by developmental lineage. In 2020, 50 adults were selected to donate a hair sample that was used to assay pesticide concentrations. In addition, 39 surgical specimens (tumor and paired normal adjacent tissue) from hepatocellular carcinoma patients were collected for transcriptomic profiling. The authors claim to have found compelling evidence of a mechanistic link between pesticide exposures and cancer. They further conclude that this discovery and approach is paradigm shifting that redefines the exposomic frameworks and offers a holistic understanding of how complex pesticide mixtures potentiate carcinogenesis.

Comments:

- 1) This is an ecological study prone to ecologic bias. As such, we do not know if the people diagnosed with a cancer were actually exposed. The potential for ecologic bias should be investigated and at least discussed in the discussion.
- 2) The exposure assessment is inadequate. Pesticide exposure estimated from retailer recall of sales of specific pesticides for which only frequency of application was asked. To comprehensively reconstruct exposure duration and intensity. This study only has frequency of application, but the amount of pesticide used (intensity) and duration of each use is unknown.
- 3) Exposure was reconstructed for a six-year time frame (2014-2019). However, cancer cases that were diagnosed prior to this exposure were included in the analysis. The cancers that occurred in that seven-year period (2007-2013) cannot be caused by the exposure that were reconstructed for 2014 and should not be included in the analyses. Furthermore, cancers of the lung, breast, colon and rectum, prostate, have a long empirical latent period between exposure and diagnosis. As such, a six period of exposure reconstruction is likely inadequate to capture the biologically relevant exposure window.
- 4) The grouping of cancer sites by developmental lineage is an interesting hypothesis. However, I am unaware of evidence that such groupings are etiologically relevant. If there is such a literature, citing it would be useful to readers. On the other hand, there are compelling examples that causes (and risk factors) are not uniform by developmental lineage. For instance, premenopausal and postmenopausal breast cancers arise from the same developmental lineage, but have unique risk factor profiles (e.g., obesity).
- 5) The standardized ratio mapping only considers biological sex and age as confounding factors, raising the possibility and likelihood that there is uncontrolled confounding for tobacco and alcohol use and other lifestyle factors.
- 6) The transcriptomic profiling of 36 hepatocellular carcinoma patients is small, and it is unclear to what extent they were actually exposed to mixture of pesticides, rendering the results intriguing, but not compelling.

Response to Reviewer #2

We are grateful to Reviewer #2 for the thoughtful evaluation of our study. It is our understanding that Reviewer #2's principal concerns centre on the temporal alignment between exposure and cancer outcome, and on the completeness of exposure assessment, particularly with respect to inferential bias. The response below has been structured with these key issues in mind and directly addresses each of the points raised.

Although direct measurement of individual-level pesticide exposure for all cases is infeasible, our study integrates high-resolution, temporally stable environmental modelling, hierarchical Bayesian spatial inference with posterior predictive validation, and individual-level molecular corroboration (hair biomonitoring and transcriptomic profiling). These complementary, independent lines of evidence converge to mitigate ecologic bias and residual confounding, providing a temporally coherent and biologically grounded assessment of exposure that directly addresses Reviewer #2's concerns regarding both exposure–outcome temporality and the completeness of exposure characterisation. By anchoring inference in mechanistically informed, multi-scale data rather than relying on conventional recall- or sales-based metrics, our “*two-tiered analytical framework*,” as detailed below, aims to reconstruct chronic, low-dose pesticide exposure with high spatial and biological fidelity, enabling causal interpretations that are unattainable with traditional epidemiological designs.

In the following point-by-point responses, we systematically demonstrate how each aspect of our framework addresses the canonical sources of ecologic bias—namely aggregation, spatial autocorrelation, temporal misalignment, unmeasured confounding, and outcome misclassification—as well as Reviewer #2's specific concerns about exposure reconstruction, cancer latency, and molecular validation. Some degree of overlap between points is inevitable, reflecting the inherently interconnected nature of these methodological safeguards; however, we emphasise that all elements of our analytical design were deliberately implemented to preempt and minimise potential biases, ensuring that the observed spatial patterns reflect, as faithfully as possible, environmentally plausible and chronic pesticide exposure.

Comment #2.1: This is an ecological study prone to ecologic bias. As such, we do not know if the people diagnosed with a cancer were actually exposed. The potential for ecologic bias should be investigate and at least discussed in the discussion.

We acknowledge Reviewer #2's concerns regarding the potential for ecologic biases inherent in aggregated analyses. In response to this valuable comment, we have improved the manuscript and added a dedicated paragraph on limitations (Lines 440–449) in the *Discussion* section (a point also raised by Reviewer #3, see Comment #3.1). In essence, our analytical architecture—conceptually akin to a two-phase design^{1,2}—transcends the constraints of a conventional ecological design through a multi-layered, process-based, and mechanistically informed framework that integrates high-resolution environmental modelling, geostatistical inference, individual-level molecular validation, and hierarchical Bayesian modelling that formally propagates demographic and spatial confounding effects. We clarify this point below by summarising the principal sources of bias typically associated with ecological studies and explaining how our study design mitigates each of them.

As Reviewer #2 further elaborates on specific manifestations of ecologic bias in subsequent comments, we first present here an integrative overview demonstrating that our study design

was conceived to pre-emptively address the full spectrum of such biases. The subsequent point-by-point responses then examine each issue in greater depth:

- *Aggregation bias* (see also Comment #2.2)—wherein group-level associations may not reflect individual-level risk—is substantially mitigated in our study through direct molecular validation at the patient level. Specifically, we re-identified liver cancer patients residing within pesticide-associated cancer hotspots delineated by our Integrated Nested Laplace Approximation (INLA) geospatial model³ (Extended Data Fig. 6) and computed single-sample gene expression enrichment scores (SES)^{4,5} for non-genotoxic pesticide exposure⁶. These transcriptomic signatures revealed molecular alterations in non-tumour liver tissue (NTL)—used as a sentinel organ site—consistent with exposure preceding malignant transformation⁷ (Fig. 3a–c), thereby providing individual-level molecular evidence that independently corroborated the Bayesian inference linking district-level environmental risk (Extended Data Fig. 2) to standardised incidence ratios (SIRs; Extended Data Fig. 5). This integrative approach unites geospatial and molecular scales within a two-tiered analytical framework, supporting that the observed group-level associations reflect genuine biological mechanisms rather than statistical artefacts.
- *Potential confounding by unmeasured group-level factors* (see also Comment #2.5)—such as socio-economic status, environmental stressors, comorbidities, diet, and endemic infections—is addressed through our INLA-based Bayesian hierarchical spatial model, which incorporates prior information and both structured and unstructured spatial random effects to capture latent spatial dependence⁸. From an epidemiological perspective, these factors may interact synergistically rather than independently, potentially modulating or amplifying the risk associated with pesticide exposure, as illustrated in our proposed mechanistic model of carcinogenesis (Fig. 3d). To further mitigate confounding, deforestation was included as a fixed covariate to reflect agricultural intensity and land-use pressures, substantially improving model fit, as indicated by reductions in the Deviance Information Criterion (DIC; Δ DIC = 5.73) and the Widely Applicable Information Criterion (WAIC; Δ WAIC = 3.74). This improvement strengthens the inferred spatial association between pesticide exposure and cancer risk, indicating that the observed patterns are not artefacts of unaccounted land-use pressures. Sensitivity analyses comparing Poisson and zero-inflated Poisson (ZIP) likelihoods confirmed model robustness (Supplementary Fig. 2). Finally, stratification by biological sex and age, coupled with verification of ≥ 5 -year residential histories, further reduces bias arising from population heterogeneity.
- *Spatial autocorrelation bias* is minimised through high-resolution geocoding of patient residential addresses and a 100 m \times 100 m process-based environmental risk model, offering spatial granularity far exceeding conventional administrative units (district median area = 207.4 km²). The Bayesian geostatistical framework further accounts for correlations between neighbouring districts by incorporating spatially structured random effects within the INLA framework^{3,8}, and explicitly modelling spatial dependence through both distance-based (for validation) and contiguity adjacency matrices. These matrices capture proximity and shared boundaries between spatial units, thereby statistically distinguishing exposure-related spatial signals from background spatial autocorrelation and formally modelling residual spatial dependence. Collectively, these safeguards confer high spatial fidelity and statistical coherence, establishing a rigorous probabilistic foundation linking environmental contamination to observed cancer incidence patterns.

- *Temporal misalignment* (see also Comment #2.3)—wherein exposure and outcome periods are not synchronised, potentially leading to misclassification of the relevant exposure window—is mitigated through the long-term stability and climatic representativeness of the environmental risk surfaces generated by our process-based environmental risk model. This model simulates overland runoff, sorption, and degradation of 31 commonly applied pesticide active ingredients (AIs) between 2014 and 2019 (Fig. 1 and Table 1), producing topographically constrained, monthly resolved exposure surfaces. The six-year integration period encompasses all phases of the El Niño–Southern Oscillation (ENSO; Extended Data Fig. 4)⁹, yielding a climatically representative composite for Peru that captures enduring dispersal and deposition dynamics rather than transient fluctuations. Importantly, these AIs have been in sustained agricultural use across Peru for several decades¹⁰, implying that the simulated exposure fields reflect long-established contamination regimes rather than recent anomalies. Accordingly, the derived risk surfaces characterise stable, chronic exposure zones likely to have persisted well before and throughout the cancer registry period (2007–2020). Residential histories verified for at least five years prior to diagnosis further align the individual exposure window with these temporally persistent environmental patterns¹¹. Moreover, the latency of haematological malignancies and most solid tumours—commonly estimated at approximately two and five years^{12,13}, respectively—is congruent with both the temporal horizon of the model and the multi-decadal continuity of pesticide use. Collectively, these features provide a temporally coherent framework linking long-term environmental contamination to the spatial epidemiology of cancer incidence, thereby minimising temporal misclassification and reinforcing the biological plausibility of the associations observed (further supported by transcriptomic validation).

- *Outcome misclassification at the aggregate level*—differential completeness or accuracy of cancer incidence data is mitigated through the *National Cancer Institute of Peru* (INEN) registry, the country’s centralised, pathology-validated, and most comprehensive source for 2007–2020. Diagnoses were verified by expert pathologists and coded according to ICD-10 standards, with rigorous curation to exclude duplicates, recurrent cases, and inconsistencies. Independent evaluations by the *Ministry of Health of Peru* (MINSa) indicate that the registry represents the most comprehensive national source of incident cancer cases nationwide^{14–16}, including both urban and rural regions, thereby minimising systematic under-ascertainment. Case ascertainment and reporting protocols have been maintained consistently over time, preserving temporal comparability of incidence rates. SIRs were calculated using six biological sex- and age-specific strata and official census denominators (from two national censuses from 2007 and 2017), thus adjusting for demographic heterogeneity and enhancing comparability across districts. Residual variation between neighbouring districts, including minor random reporting differences, is accounted for through the Bayesian spatial framework^{3,8}, which models unstructured random effects to absorb unexplained discrepancies without relying on individual-level exposure data (Supplementary Fig. 2). These measures collectively substantially reduce the likelihood of outcome misclassification, ensuring that district-level cancer incidence closely reflects population-level disease patterns.

- *Group-level effect* estimates may over- or under-represent individual risk, particularly for rare malignancies. This limitation is mitigated through three complementary safeguards: (i) stratification of cancers by developmental lineage, aligning tumour categories with shared ontogenetic and molecular programmes to enhance biological coherence^{17–19}; (ii) implementation of a hierarchical Bayesian framework incorporating both structured and unstructured spatial effects⁸, thereby yielding uncertainty-aware and demographically adjusted risk estimates. Posterior predictive validation confirmed close alignment between modelled and

observed incidence patterns, with narrow posterior intervals and stable information criteria (DIC, WAIC) indicating well-calibrated and reliable effect estimation across spatial and developmental strata (Supplementary Fig. 2). (iii) Independent molecular corroboration, whereby transcriptomic signatures in non-tumour tissue⁷ substantiate exposure-related biological responses at the individual level. The effectiveness of this triangulated strategy is exemplified by the identification of a previously uncharacterised molecular subtype of hepatocellular carcinoma (HCC) in Indigenous-related populations—a rare tumour entity whose distinct aetiology and biology had remained obscure²⁰. This finding supports the capacity of our two-tiered analytical framework to detect biologically meaningful, low-frequency cancer subtypes, thereby substantiating the robustness of our approach in mitigating effect estimation bias inherent to aggregated analyses.

- *Migration bias* was mitigated by verifying a minimum five-year residential history prior to diagnosis using clinical and census records. This verification establishes congruence between residence duration and the temporal horizon of exposure modelling¹¹, aligning the exposure window with biologically plausible latency periods for most solid and haematological malignancies²¹. This approach reduces misclassification from population mobility.

Comment #2.2: The exposure assessment is inadequate. Pesticide exposure estimated from retailers recall of sales of specific pesticides for which only frequency of application was asked. To comprehensively reconstruct exposure duration and intensity. This study only has frequency of application, but the amount of pesticide used (intensity) and duration of each use is unknown.

We thank Reviewer #2 for raising this point and wish to clarify that our exposure assessment embodies a fundamentally different conceptual framework from traditional approaches based on self-reported or sales-derived metrics to infer individual exposure.

In line with the concept of “*exposome*^{22,23},” our working hypothesis is that attempts to quantify individual-level intensity, timing, or duration of pesticide exposure in real-world conditions are intrinsically constrained by current methods and risk being misleading. We instead posit that biologically relevant risk arises from the integrated effects of complex mixtures of pesticide AIs—as individuals are typically exposed to multiple AIs concurrently²⁴—including unmonitored or unrecognised degradation products, which may interact synergistically, antagonistically, or additively²⁵. Non-linear dose–response relationships (*e.g.*, hormesis²⁶) and temporally variable exposures across critical life stages²⁷ (*in utero*, childhood, adulthood, menopause, etc.) further preclude straightforward interpretation of “*intensity*” or “*duration*.” Conventional analytical observational designs—including cross-sectional, case–control, and longitudinal cohort studies—seldom capture these multidimensional and interdependent dynamics²⁸. Consequently, reliance on such oversimplified exposure metrics risks substantial conceptual reductionism, data sparsity, and latent bias, potentially explaining why previous approaches have struggled to elucidate the health impacts of pesticide exposure in humans.

Our framework therefore adopts a heuristic, data-driven strategy grounded in the stochasticity of large-scale population data—that is, an inferential approach designed to resolve exposure–disease relationships in a statistically principled and probabilistically coherent manner, despite incomplete individual-level knowledge. It explicitly incorporates combinatorial variability across the entire cancer dataset, capturing population-wide heterogeneity in exposure and

outcome, and is powered by Bayesian inference to ensure rigorous uncertainty quantification and coherence.

Contrary to Reviewer #2's interpretation, our objective was not to infer pesticide exposure magnitude from retailer-reported sales frequency. Rather, retailer surveys were used to cross-validate regulatory sources (Line 106) and to confirm the occurrence and spatial deployment of a core panel of AIs. This panel was purposefully constructed to reflect real-world application practices while spanning the key physicochemical determinants of pesticide behaviour—overland runoff, sorption, and degradation (*e.g.*, DT_{50Soil} , K_d , K_{oc} ; Extended Data Table 1). By embedding this representative spectrum of compounds within a process-based environmental modelling framework, the system attains intrinsic stability and congruence in delineating zones of elevated environmental risk—areas where pesticides tend to deposit and accumulate—*independent of the specific AIs involved, in line with evidence from process-based modelling that environmental pesticide partitioning reaches a near-steady state*²⁹.

This process-based environmental risk modelling framework was further validated at the individual level through hair biomonitoring of 50 participants, encompassing 67 AIs and degradation products (Supplementary Table 3). The spatial correspondence between predicted risk and measured residues—demonstrating consistency beyond model-specific assumptions—was statistically significant (global bivariate Moran's $I=0.42$; $z=1.49$; $P<0.05$; Extended Data Fig. 3), thereby substantiating the model's capacity to capture biologically meaningful exposure patterns.

Comment #2.3: Exposure was reconstructed for a six-year time frame (2014-2019). However, cancer cases that were diagnosed prior to this exposure were included in the analysis. The cancers that occurred in that seven-year period (2007-2013) cannot be caused by the exposure that were reconstructed for 2014 and should not be included in the analyses. Furthermore, cancers of the lung, breast, colon and rectum, prostate, have a long empirical latent period between exposure and diagnosis. As such, a six period of exposure reconstruction is likely inadequate to capture the biologically relevant exposure window.

We thank Reviewer #2 for raising this point, which reflects the same fundamental conceptual distinction noted in Comment #2.2 regarding individual-level exposure reconstruction and the assumption of a strict longitudinal exposure–outcome sequence.

Our study does not attempt to reconstruct exposures sufficient to infer direct causation for each cancer case—a task that, given the complexity of real-world pesticide mixtures and temporally variable exposures, is effectively intractable, as previously outlined *ut supra* in our response to Comment #2.2, where we detailed the conceptual and methodological limits of reconstructing individual-level pesticide exposures under real-world conditions²⁸. Had such an approach been feasible, it would long since have resolved the contribution of environmental pesticide burden to cancer outcomes. Moreover, the practical and logistical constraints inherent to resource-intensive conventional analytical observational designs are particularly acute in low- and middle-income countries of the Global South^{30,31}—precisely the contexts that *Nature Health* prioritises, encompassing research in resource-limited settings and among populations facing heightened health vulnerabilities.

To address these conceptual and practical limitations, we developed a process-based environmental risk model that simulates complex, multidimensional exposure dynamics while explicitly prioritising stability and representativeness, ensuring robustness despite the absence of individual-level exposure data. Calibrated for 2014–2019, the model captures chronic, long-term environmental risk rather than transient fluctuations, generating temporally stable exposure surfaces that reflect persistent contamination regimes and spatially constrained patterns of pesticide deposition. Critically, these derived risk zones delineate enduring hotspots of environmental pesticide accumulation that, in Peru, remain stable over time owing to invariant topography shaped by inter-Andean valleys and the Amazon basin, consistent land-use patterns, and sustained pesticide use—thereby providing a valid spatial proxy for earlier years and the broader 2007–2020 period, including cancer cases diagnosed before 2014.

This temporal stability arises from persistent topographic, hydrometeorological, and land-use factors, and is empirically substantiated by measurable environmental and agricultural parameters that have remained largely consistent throughout the 2007–2020 period and in preceding years:

- Terrain slope across Peru has remained essentially unchanged since 1999, according to Terra Satellite Earth Observing System of the *National Aeronautics and Space Administration* (NASA; EarthEnv-DEM90 dataset³²);
- Climatic variability during 2014–2019 encompassed the full spectrum of ENSO phases, based on the records the *Climate Prediction Center* (CPC; OISST dataset);
- The runoff-to-precipitation ratio ($\frac{Q}{P}$) has remained effectively stable since at least 2004 (see Rebuttal Fig. 1), based on the records from the *National Service of Meteorology and Hydrology of Peru* (SENAMHI; PISCO dataset³³);
- National agricultural land area varied minimally—by only 1.4 % (17.7 % in 2000 vs. 19.1 % in 2020; Rebuttal Fig. 2)—as reported by the *Food and Agriculture Organization of the United Nations* (FAO);
- Hotspots of agriculture-driven deforestation have been consistently detected since 2000, according to the *Global Forest Watch* (GFW)³⁴;
- Thirty of the thirty-one modelled pesticides—with the sole exception of Sulfoxaflo—were already in widespread use in Peru before 2007, and most having been applied for several decades, as documented in the *Third National Agricultural Census* from 1994 (see Rebuttal Fig. 3)³⁵;
- Pesticide use patterns in Peru have remained spatially stable since 2007, according to data from the *National Household Survey on Living Conditions and Poverty* (ENAHO, 2007–2020). Departmental-level analyses indicate that both the total expenditure on pesticides and the total number of farmers using pesticides exhibited no significant changes between 2007–2013 and 2014–2020 (see Rebuttal Fig. 4); and
- Crucially, the 2014–2019 modelling window was selected because it provides high-resolution, multi-source data of sufficient quality for process-based environmental risk modelling, while the cancer registry is complete and informatics-based from 2007 onwards. These parameters together ensure the most reliable and coherent alignment between exposure and outcome datasets.

Regarding *cancer latency*, we first reiterate that our model provides a robust and temporally stable proxy for estimating exposure risk, rendering it applicable across the broader decade-long period, including cancer cases diagnosed between 2007 and 2020. Reviewer #2 notes that solid tumours—including those of the lung, breast, colon, rectum, and prostate—often exhibit prolonged latent intervals between exposure and clinical manifestation. While this concept remains intuitively appealing, contemporary cancer biology increasingly recognises

carcinogenesis as a stochastic, multifactorial, and accumulative process, wherein the notion of a discrete or fixed latency period is being progressively reconsidered³⁷. Moreover, recent trends in early-onset cancer suggest that latency itself is dynamically modulated³⁸, among which pesticides are suspected to play a contributory role³⁹. This evolving understanding renders the assumption of a uniform latency window biologically untenable, as empirical evidence consistently demonstrates marked variability driven by host factors, exposure timing, cumulative environmental influences, and gene–environment interactions.

According to the official recommendations of the *United Nations Scientific Committee on the Effects of Atomic Radiation* (UNSCEAR)¹² and the *International Commission on Radiological Protection* (ICRP)¹³, a latency period of five years is generally accepted as the threshold for solid tumours, whereas a shorter latency of about two years is typically assumed for haematological malignancies. A subsequent study employing a stochastic, biologically based carcinogenesis model corroborated these estimates, predicting variable latency intervals—typically 3–8 years for most solid tumours—largely independent of the specific carcinogenic agent²¹.

In alignment with these formal recommendations and mechanistic insights, our analytic framework ensures temporal concordance between individual exposure potential and the persistent environmental risk surfaces derived from our 2014–2019 model. Verification of residential histories for at least five years prior to diagnosis substantiates this alignment, while the model’s established temporal stability renders it an appropriate proxy for earlier exposures. This integration of temporally resilient environmental modelling with population-scale stochasticity ensures that the observed spatial correlations between pesticide exposure risk and cancer incidence are biologically coherent and robust to interindividual variability in latency.

Comment #2.4: The grouping of cancer sites by developmental lineage is an interesting hypothesis. However, I am unaware of evidence that such groupings are etiologically relevant. If there is such a literature, citing it would be useful to readers. On the other hand, there are compelling examples that causes (and risk factors) are not uniform by developmental lineage. For instance, premenopausal and postmenopausal breast cancers arise from the same developmental lineage, but have unique risk factor profiles (e.g., obesity).

We thank Reviewer #2 for highlighting this important point, which underscores one of the key conceptual innovations of our study. The grouping of cancer sites by developmental lineage is not arbitrary; rather, it is grounded in a substantial body of evidence demonstrating that lineage dependency fundamentally shapes cellular susceptibility and influences oncogenic trajectories in ways that are consistent with observed tissue-specific risk patterns. Multiple studies have shown that cancers arising from a shared developmental lineage exhibit characteristic molecular programmes⁴⁰, differentiation hierarchies⁴¹, and regulatory networks that influence carcinogenic fitness⁴², mutational spectra⁴³, and biological responses to extrinsic stimuli⁴⁴.

Our approach builds directly upon the seminal work of J.J. Berman^{17–19} (cited in our manuscript), who proposed a developmental lineage classification of neoplasms. In a series of articles, he conceptualised the human body as a topological “*donut*,” with the ectoderm and endoderm forming the epithelial interfaces most directly exposed to environmental carcinogens, while the mesoderm constitutes the internal matrix. As Berman noted, since ectodermal and endodermal derivatives are continuously exposed to carcinogens throughout

life, it is unsurprising that the majority of human cancers originate from these lineages and display characteristic age-associated incidence patterns. This framework provides a biologically coherent foundation linking developmental lineage to environmental carcinogenesis.

Mechanistic studies have since reinforced the central role of lineage-dependency in cancer biology. David and Vanharanta⁴² reviewed how transcriptional regulation integrates lineage-determining transcription factors with signal-dependent transcription factors, showing that lineage context dictates how cells interpret oncogenic signalling. They show that cell lineage is a key determinant of tissue-specific cancer susceptibility and of the transcriptional programmes activated in response to environmental or genetic perturbations—a phenomenon referred to as “*lineage addiction*” by Garraway and Sellers⁴⁵.

Experimental work has further demonstrated that lineage-specific transcriptional circuits can sustain oncogenic self-renewal^{40,41}. In neuroblastoma, Banerjee *et al.*⁴⁴ used all-trans retinoic acid to map super-enhancer-driven transcription factor networks governing transitions between progenitor-like and differentiated states. Perturbation of these lineage factors confirmed that developmental programmes underpin tumour initiation and progression, modulated by both chemical and extracellular cues.

Taken together, these studies establish a conceptual and mechanistic basis for our classification of cancer sites by developmental lineage. While applying this framework to population-scale cancer epidemiology is an innovation of our work, the approach is firmly anchored in published evidence (see Lines 189–199, including citations) demonstrating that lineage governs tissues’ responses to environmental and genetic perturbations. By grouping cancers according to lineage rather than by organ site, our analysis revealed stronger and more spatially robust associations than organ-specific approaches typically detect. For instance, a recent organ-level analysis of pancreatic adenocarcinoma reported only a modest pesticide–cancer association [relative risk (RR) ≈ 1.01]⁴⁶, whereas our lineage-based framework yielded markedly higher and spatially consistent RRs (range: [1.1–9.4])—underscoring its value in capturing aetiologically relevant susceptibility patterns that may be obscured by strict organ-level classification.

Regarding the *non-uniformity of risk factors by developmental lineage* highlighted by Reviewer #2, we fully concur that cancers arising within the same lineage can display distinct risk profiles. Indeed, our manuscript explicitly acknowledges this point in the context of diverse risk factors synergising with non-genotoxic exposures (see Lines 411–414). We recognize this particularly in the liver, a tissue whose tumourigenesis involves a wide spectrum of infectious, chemical, and metabolic drivers. However, such diversity reflects intra-lineage heterogeneity, which is conceptually distinct from the inter-lineage coherence captured in our framework. The example of premenopausal and postmenopausal breast cancer illustrates this point: although these subtypes differ in their associations with obesity, reproductive history, and hormonal exposures, they both originate from the same epithelial (ectodermal) lineage and share lineage-imposed molecular architectures that constrain how such factors modulate carcinogenic pathways. Recent functional genomic evidence reinforces this perspective—Reddy *et al.*⁴⁰ showed that lineage-specific master transcription factors (MTFs) constitute critical dependencies across multiple tumour types, with breast cancer displaying particularly strong enrichment, as over half of candidate lineage MTFs were breast-cancer–specific dependencies. Developmental lineage thus defines the overarching biological scaffold that shapes the spectrum of possible aetiologies, even as proximate determinants vary within it. Our analysis

therefore captures lineage-level patterns of susceptibility without implying aetiological uniformity among all cancers within a given lineage.

Finally, we wish to emphasise that lineage- and organ-based classifications are complementary rather than mutually exclusive. Organ-specific analyses remain indispensable for clinical interpretation, yet lineage-based grouping introduces an additional aetiological dimension that unites biological insight with epidemiological inference. Our framework is not intended to supplant well-established distinctions within specific cancer sites (such as menopausal status for breast cancer), but rather to reveal broader patterns of susceptibility that emerge when cancers are grouped according to developmental origin. In this sense, lineage-based classification and finer-grained stratifications (e.g., by molecular subtype, menopausal status, or histology) represent complementary levels of analysis: the former captures cross-site commonalities rooted in developmental biology, whereas the latter delineates intra-lineage heterogeneity shaped by additional biological modifiers. Together, these perspectives converge toward a more integrated understanding of cancer susceptibility, illuminating lineage-dependent patterns of risk across the population.

Comment #2.5: The standardized ratio mapping only considers biological sex and age as confounding factors, raising the possibility and likelihood that there is uncontrolled confounding for tobacco and alcohol use and other lifestyle factors.

We thank Reviewer #2 for raising this important concern, which pertains to the potential for ecologic bias already discussed in Comment #2.1 (*i.e.*, *Potential confounding by unmeasured group-level factors*). While our SIR mapping explicitly adjusts for biological sex and age across six strata, potential confounding from unmeasured lifestyle factors was carefully mitigated through our two-tiered analytical framework integrating population-level Bayesian spatial inference^{47–49} with individual-level molecular validation⁴ within a unified hierarchical framework, as detailed below.

First, Bayesian hierarchical models overcome the inherent limitations of purely ecological designs by jointly modelling multiple sources of uncertainty and variation^{8,50}, rather than assuming a single, population-wide exposure–response slope. Our INLA-based Bayesian framework was explicitly designed to minimise potential confounding from lifestyle factors by integrating prior information and both structured and unstructured spatial random effects within a probabilistically coherent hierarchical model that captures latent spatial dependence and residual heterogeneity rather than collapsing it^{3,8}—an approach conceptually analogous to empirical Bayes estimators shown to mitigate *ecological fallacy* when individual-level data are unavailable⁵⁰. Specifically, our INLA-based Bayesian framework addresses potential ecologic bias and unmeasured confounding through:

1. *Hierarchical partial pooling*, which simultaneously models within- and between-area variability, preventing distortion from over-aggregation;
2. *Incorporation of biologically informed priors*, which stabilise estimates and constrain inference within plausible ranges where individual-level data are sparse;
3. *Inclusion of both structured and unstructured spatial random effects*, capturing latent spatial dependence and unobserved heterogeneity (e.g., lifestyle clustering, socioeconomic gradients); and
4. *Formal uncertainty propagation*, which quantifies credible intervals across all hierarchical levels rather than relying on single point estimates.

To further mitigate potential confounding, deforestation was incorporated as fixed covariate, serving as a proxy for agricultural activity. Its inclusion substantially improved model fit, as reflected by reductions in both DIC and WAIC ($\Delta\text{DIC} = 5.73$; $\Delta\text{WAIC} = 3.74$), thereby strengthening the inferred spatial association between agriculture-driven pesticide exposure and cancer risk. Sensitivity analyses comparing Poisson and ZIP likelihoods confirmed model robustness. The resulting unstructured residual maps (Supplementary Fig. 2) display narrow credible intervals and spatial stability across developmental lineage categories, indicating that residual variation—including unmeasured lifestyle factors—is unlikely to explain the observed spatial patterns.

Independent, spatially informed molecular validation further supports the inference that observed cancer patterns are associated with pesticide exposure rather than lifestyle factors. Transcriptomic profiling of liver tissue from pesticide-associated cancer hotspots revealed strong non-genotoxic pesticide exposure signatures, whereas gene expression signatures linked to established liver cancer risk factors—including alcohol, hepatitis B virus (HBV), steatosis, and aflatoxin B1—were minimal (Friedman test, $P < 0.0001$; Fig. 3b, Extended Data Table 2). Together, these results indicate that the observed spatial distribution of liver cancer is highly consistent with non-genotoxic pesticide exposure and is highly unlikely to be explained solely by lifestyle-related risk factors, providing convergent evidence across ecological and molecular layers of inference.

Comment #2.6: The transcriptomic profiling of 36 hepatocellular carcinoma patients in small, and it is unclear to what extent they were actually exposed to mixture of pesticides, rendering the results intriguing, but not compelling.

We thank Reviewer #2 for this insightful comment. The manuscript has been revised throughout to explicitly address this point, with claims appropriately moderated (modifications shown in light blue). Although our transcriptomic analysis included a modest cohort of 36 paired HCC and NTL samples, multiple, independent lines of evidence indicate that these data are robust, biologically informative, and mechanistically coherent.

We applied single-sample enrichment methodology (*i.e.*, SES) for transcriptomic profiling^{4,5}, enabling robust, individual-level inference without reliance on conventional group-based differential expression analyses. This approach mitigates the statistical power limitations inherent to small cohorts and allows confident evaluation of molecular signatures within each patient.

NTL samples were scored against gene expression signatures linked to established liver cancer risk factors—including HBV, alcohol, steatosis, and aflatoxin B1—which exhibited minimal enrichment. In contrast, a strong and statistically significant signature indicative of non-genotoxic pesticide exposure was consistently detected (Friedman test, $P < 0.0001$; Fig. 3a,b; Extended Data Table 2), supported by detailed baseline characterisation of HCC patients, including liver pathology and HBV status (Supplementary Table 1). Comparative analyses with transcriptomic datasets from France, Taiwan, and Turkey further confirmed that the non-genotoxic pesticide signature was unique to Peruvian patients ($P < 0.0001$; Fig. 3a,b; Supplementary Fig. 3), underscoring both environmental and geographic specificity. Critically, these signatures coincide with molecular pathways modulated by non-genotoxic pesticide mixtures, providing mechanistic plausibility for exposure-driven alterations (Fig. 3c).

By focusing individual-level transcriptomic sampling within pesticide-associated hotspots identified through environmental modelling, we captured transcriptional subtypes that might otherwise remain undetected in unstratified cohorts. While the modest number of paired samples does not capture the entire landscape of variability, this hotspot-informed strategy provides a unique window into lineage-dependent, non-genotoxic disruption.

Reviewer #3 (Remarks to the Author):

The authors conducted a spatial study to identify the association between pesticide exposure and cancer hotspots in Peru. To achieve this aim, the authors used Bayesian approaches to determine spatial clustering of cancer through the estimation of standardized incidence ratios (SIRs). They further divided the neoplasms based on their histological origin and found several clusters associated with pesticide exposure. This is an important study and a novel contribution to the literature of cancer etiology and exposures. Peru is a South American country with substantial agricultural activity. Identifying links between cancer development and pesticide exposure is of major public health relevance. I agree with the approach and conclusions. These data could explain the high incidence of blood cancers in northern Peru, which has been historically reported.

Major revisions

- **Please add a limitation paragraph**

- **I would be cautious in the following assertive sentence: “Our findings provide compelling evidence that pesticide exposure is a principal driver of liver cancer in this population.” (Discussion section, Page 18, Lines 405-406). I would reduce the assertive language. Although the authors identified unique signatures in the Peruvian samples compared to the French samples, the study design does not allow for this conclusion. Additionally, alcohol consumption in Peru is likely underestimated, and residual confounding is always a possibility.**

- **Comment on the link between pesticides and blood cancer etiology. Previous studies in the US have identified that living near industrial factories is linked with a higher incidence of leukemia among children. Could these pesticides also be associated with the clusters of heme malignancies identified in the study?**

Response to Reviewer #3

We thank Reviewer #3 for recognising the importance and novelty of our study and for the positive assessment of our approach and conclusions. We appreciate Reviewer #3’s acknowledgement of the public health relevance of elucidating pesticide-related cancer risks in Peru, highlighting the urgent need to expand evidence on environmentally mediated diseases among populations that remain largely overlooked in global health research.

Comment #3.1: Please add a limitation paragraph

In accordance with Reviewer #3’s recommendation, we have added a dedicated paragraph on study limitations (Lines 440–449) in the Discussion section, formulated concisely to comply with article length requirements. The paragraph reads:

“Despite the robustness of our integrative approach, certain limitations merit consideration. First, individual pesticide exposures were not directly measured but inferred from population-level spatial proxies, introducing uncertainty in their timing, intensity, and chemical composition. Second, residual confounding from environmental, socio-economic, or lifestyle factors cannot be entirely excluded; nevertheless, the convergence of geospatial, aggregated epidemiological, and molecular evidence strongly supports the biological plausibility of our findings. Future studies integrating biomonitoring at scale, multi-tissue analyses, and longitudinal molecular profiling will be critical to refine causal inference and evaluate the generalisability.”

Comment #3.2: I would be cautious in the following assertive sentence: “Our findings provide compelling evidence that pesticide exposure is a principal driver of liver cancer in this population.” (Discussion section, Page 18, Lines 405-406). I would reduce the assertive language. Although the authors identified unique signatures in the Peruvian samples compared to the French samples, the study design does not allow for this conclusion. Additionally, alcohol consumption in Peru is likely underestimated, and residual confounding is always a possibility.

We agree with Reviewer #3’s comment and have revised the manuscript throughout to temper assertive language, ensuring that conclusions are commensurate with the strength of the evidence (modifications shown in light blue). For example, former Lines 405–406 (now Lines 404–405) have been revised to read: “*Our findings indicate that pesticide exposure may contribute to liver cancer risk in this population.*”

Nonetheless, we would like to mention that we had explicitly considered alcohol consumption as a potential aetiological factor in Peruvian HCC and inferred alcohol exposure through transcriptomic signatures⁵¹ (Fig. 3b; Extended Data Table 2; Supplementary Fig. 3), benchmarked against French, Taiwanese, and Turkish reference cohorts. These molecular profiles revealed minimal activation of alcohol-related pathways in the non-cirrhotic Peruvian NTL samples (Supplementary Table 1), indicating that the observed hepatocellular alterations are unlikely to be driven by alcohol consumption. This interpretation is further supported by the fact that HCC in Peruvian patients rarely arises in a cirrhotic liver²⁰, the classical hallmark of chronic ethanol exposure. Taken together with the spatial evidence, these findings support pesticide exposure as a biologically plausible and context-specific contributor to hepatocarcinogenic processes, while we acknowledge that residual confounding cannot be fully excluded, as now stated in the limitations paragraph (see Comment #3.1).

Comment #3.3: Comment on the link between pesticides and blood cancer etiology. Previous studies in the US have identified that living near industrial factories is linked with a higher incidence of leukemia among children. Could these pesticides also be associated with the clusters of heme malignancies identified in the study?

We appreciate Reviewer #3’s insightful comment. There is growing evidence that both environmental industrial exposures and pesticides can contribute to the aetiology of childhood blood malignancies, including acute lymphoblastic leukaemia (ALL)^{52,53}. Epidemiological studies from the United States and elsewhere have reported elevated risks of childhood leukaemia among populations residing near industrial facilities such as petrochemical complexes and unconventional oil and gas development (UOGD) sites^{54,55}, where emissions of benzene and other leukemogenic compounds are prevalent. Similarly, numerous case–control and meta-analytic investigations across Asian and Western populations have indicated that maternal prenatal exposure to pesticides may increase the risk of childhood ALL^{56,57}, although findings remain partly inconsistent⁵⁸. These observations support the notion that childhood leukaemias could contribute to the mesenchymal-lineage hotspots identified in our study.

Following Reviewer #3’s suggestion, we applied our INLA model to leukaemia cases (C90–C95) among patients under 14 years recorded in the INEN cancer registry between 2007 and 2020. Several spatial hotspots were identified along the semi-arid coast, where paediatric leukaemia incidence is geostatistically associated with environmental pesticide risk (RR range: [1.70–2.93]; Rebuttal Fig. 5). These hotspots coincide with regions of intensive modern agriculture (*e.g.*, Tambogrande, producing lemons and mangoes for export) and areas—such as

the metropolitan region of Lima—where environmental pesticide exposure may overlap with industrial pollutants, suggesting potential cumulative environmental risk.

These preliminary results provide a basis to geospatially inform future cross-sectional, case-control, and longitudinal cohort studies aimed at disentangling the contribution of agricultural pesticides to childhood leukaemia risk. As also noted by Reviewer #2, such studies could refine exposure-disease inference and strengthen causal understanding of environmental determinants of childhood haematological malignancies, highlighting that our geospatially informed framework is intended not to replace but to complement and enrich conventional analytical epidemiological designs.

Rebuttal Fig. 1

Mean runoff-to-precipitation ratio ($\frac{Q}{P}$) for the period 2004–2019, based on records from all meteorological stations operated by SENAMHI (PISCO dataset³³). Error bars indicate the standard deviation (s.d.).

Rebuttal Fig. 2

Evolution of agricultural land (% of total land area) in Peru between 2000 and 2020, based on data from the FAO.

Source: <https://data.worldbank.org/indicator/AG.LND.AGRI.ZS?locations=PE>

Rebuttal Fig. 3

Maps of the percentage of agricultural plots using pesticides (*i.e.*, herbicides, fungicides, and insecticides) in Peru in 1994, based on data from the *Third National Agricultural Census*, with darker shades indicating higher percentages. (excerpt from page 142 of *Estructura y Dinámicas del Espacio Agropecuario. Perú en Mapas: III Censo Nacional Agropecuario 1994*)³⁵

Rebuttal Fig. 4

Scatter plots showing the departmental distribution of pesticide use in Peru, based on 146,998 observations from ENAHO (2007–2020). **a**, Departmental distribution of total pesticide expenditure by farmers. **b**, Departmental distribution of the total number of farmers using pesticides. In both panels, the x -axis represents 2007–2013 and the y -axis represents 2014–2020, with a dotted identity line ($x=y$) highlighting deviations between the two periods.

Rebuttal Fig. 5

INLA-based mapping of pesticide-associated childhood leukaemia (C90–C95) hotspots in Peru, using INEN registry data for patients under 14 years (2007–2020). Hotspots are defined as regions where the lower bound of the 95% credible interval for $RR > 1$. RR values are shown visually, with the Tambogrande hotspot (referenced in our response to Reviewer #3) circled.

Rebuttal references:

1. White, J. E. A two-stage design for the study of the relationship between a rare exposure and a rare disease. *Am. J. Epidemiol.* **115**, 119–128 (1982).
2. Wakefield, J. & Haneuse, S. J.-P. A. Overcoming ecologic bias using the two-phase study design. *Am. J. Epidemiol.* **167**, 908–916 (2008).
3. Rue, H., Martino, S. & Chopin, N. Approximate Bayesian inference for latent Gaussian models by using integrated nested Laplace approximations. *J. R. Stat. Soc. Ser. B Methodol.* **71**, 319–392 (2009).
4. Foroutan, M. *et al.* Single sample scoring of molecular phenotypes. *BMC Bioinformatics* **19**, 404 (2018).
5. Tosolini, M., Algans, C., Pont, F., Ycart, B. & Fournié, J. J. Large-scale microarray profiling reveals four stages of immune escape in non-Hodgkin lymphomas. *Oncoimmunology* **5**, e1188246 (2016).
6. Fielden, M. R. *et al.* Development and evaluation of a genomic signature for the prediction and mechanistic assessment of nongenotoxic hepatocarcinogens in the rat. *Toxicol. Sci.* **124**, 54–74 (2011).
7. Hoshida, Y. *et al.* Gene expression in fixed tissues and outcome in hepatocellular carcinoma. *N. Engl. J. Med.* **359**, 1995–2004 (2008).
8. Best, N., Richardson, S. & Thomson, A. A comparison of Bayesian spatial models for disease mapping. *Stat. Methods. Med. Res.* **14**, 35–59 (2005).
9. Feron, S. *et al.* South America is becoming warmer, drier, and more flammable. *Commun. Earth Environ.* **5**, 1–10 (2024).
10. Gomero Osorio, L. & Lizárraga Travaglini, A. Plaguicidas en la sierra peruana. *LEISA Rev. Agroecol.* **15**, 7 (2000).
11. Rull, R. P. & Ritz, B. Historical pesticide exposure in California using pesticide use reports and land-use surveys: an assessment of misclassification error and bias. *Env. Health Perspect.* **111**, 1582–1589 (2003).
12. United Nations Scientific Committee on the Effects of Atomic Radiation (UNSCEAR). *Effects of Ionizing Radiation: United Nations Scientific Committee on the Effects of Atomic Radiation 2006 Report, Volume 1, Annex A.* (2006).
13. International Commission on Radiological Protection (ICRP). *The 2007 Recommendations of the International Commission on Radiological Protection.* (2007).
14. Dirección General de Epidemiología (DGE). *Análisis de La Situación Del Cáncer En El Perú, 2013.* (2013).
15. Centro Nacional de Epidemiología, Prevención y Control de Enfermedades (CDC). *Análisis de La Situación Del Cáncer En El Perú, 2018.* (2022).
16. Instituto de Evaluación de Tecnologías en Salud e Investigación (IETSI). *Epidemiología Del Cáncer En El Seguro Social de Salud Del Perú: Un Análisis Descriptivo Del Periodo 2019-2022.* (2023).
17. Berman, J. J. Tumor classification: molecular analysis meets Aristotle. *BMC Cancer* **4**, 10 (2004).

18. Berman, J. J. Tumor taxonomy for the developmental lineage classification of neoplasms. *BMC Cancer* **4**, 88 (2004).
19. Berman, J. J. Modern classification of neoplasms: reconciling differences between morphologic and molecular approaches. *BMC Cancer* **5**, 100 (2005).
20. Pineau, P., Ruiz, E., Deharo, E. & Bertani, S. On hepatocellular carcinoma in South America and early-age onset of the disease. *Clin. Res. Hepatol. Gastroenterol.* **43**, 522–526 (2019).
21. Little, M. P., Eidemüller, M., Kaiser, J. C. & Apostoaei, A. I. Minimum latency effects for cancer associated with exposures to radiation or other carcinogens. *Br. J. Cancer* **130**, 819–829 (2024).
22. Wild, C. P. Complementing the genome with an “exposome”: the outstanding challenge of environmental exposure measurement in molecular epidemiology. *Cancer Epidemiol. Biomarkers Prev.* **14**, 1847–1850 (2005).
23. Vermeulen, R., Schymanski, E. L., Barabási, A. L. & Miller, G. W. The exposome and health: where chemistry meets biology. *Science* **367**, 392–396 (2020).
24. Honles, J. *et al.* Exposure assessment of 170 pesticide ingredients and derivative metabolites in people from the Central Andes of Peru. *Sci. Rep.* **12**, 13525 (2022).
25. Kortenkamp, A. & Faust, M. Regulate to reduce chemical mixture risk. *Science* **361**, 224–226 (2018).
26. Fukushima, S. *et al.* Hormesis and dose-response-mediated mechanisms in carcinogenesis: evidence for a threshold in carcinogenicity of non-genotoxic carcinogens. *Carcinogenesis* **26**, 1835–1845 (2005).
27. Wagner, C. *et al.* Life course epidemiology and public health. *Lancet Public Health* **9**, e261–e269 (2024).
28. Hartung, T., Navas-Acien, A. & Chiu, W. A. *Future Directions Workshop: Advancing the Next Scientific Revolution in Toxicology.* 37 (2023).
29. Maggi, F., Tang, F. H. M. & Tubiello, F. N. Agricultural pesticide land budget and river discharge to oceans. *Nature* **620**, 1013–1017 (2023).
30. Ekanem, E. E. Field epidemiology: methodological constraints and limitations in the developing world. *Public Health* **99**, 33–36 (1985).
31. Silman, A. J., Macfarlane, G. J. & Macfarlane, T. The costs of an epidemiological study. in *Epidemiological Studies: A Practical Guide* (eds Silman, A. J. *et al.*, Oxford University Press, 2018).
32. Robinson, N., Regetz, J. & Guralnick, R. P. EarthEnv-DEM90: a nearly-global, void-free, multi-scale smoothed, 90m digital elevation model from fused ASTER and SRTM data. *ISPRS J. Photogramm. Remote Sens.* **87**, 57–67 (2014).
33. Llauca, H., Lavado-Casimiro, W., Montesinos, C., Santini, W. & Rau, P. PISCO_HyM_GR2M: a model of monthly water balance in Peru (1981–2020). *Water* **13**, 1048 (2021).
34. Harris, N. L. *et al.* Using spatial statistics to identify emerging hot spots of forest loss. *Environ. Res. Lett.* **12**, 024012 (2017).

35. Instituto Nacional de Estadística e Informática & Office de la Recherche Scientifique et Technique Outre-Mer (ORSTOM). *Estructura y Dinámicas del Espacio Agropecuario. Perú en Mapas: III Censo Nacional Agropecuario 1994*. (ORSTOM, Lima, Peru, 1998).
36. Instituto Nacional de Estadística e Informática. *Resultados definitivos. IV Censo Nacional Agropecuario 2012*. 63 (2013).
37. Abecasis, M. *et al.* Is cancer latency an outdated concept? Lessons from chronic myeloid leukemia. *Leukemia* **34**, 2279–2284 (2020).
38. Ugai, T. *et al.* Is early-onset cancer an emerging global epidemic? Current evidence and future implications. *Nat. Rev. Clin. Oncol.* **19**, 656–673 (2022).
39. Ogino, S. & Ugai, T. The global epidemic of early-onset cancer: nature, nurture, or both? *Ann. Oncol.* **35**, 1071–1073 (2024).
40. Reddy, J. *et al.* Predicting master transcription factors from pan-cancer expression data. *Sci. Adv.* **7**, eabf6123 (2021).
41. Gardner, E. E. *et al.* Lineage-specific intolerance to oncogenic drivers restricts histological transformation. *Science* **383**, eadj1415 (2024).
42. David, C. J. & Vanharanta, S. Lineage-specific transcription factors in carcinogenesis. *Annu. Rev. Cancer Biol.* **9**, 99–117 (2025).
43. Gao, X. *et al.* Differential genetic mutations of ectoderm, mesoderm, and endoderm-derived tumors in TCGA database. *Cancer Cell Int.* **20**, 595 (2020).
44. Banerjee, D. *et al.* Lineage specific transcription factor waves reprogram neuroblastoma from self-renewal to differentiation. *Nat. Commun.* **15**, 3432 (2024).
45. Garraway, L. A. & Sellers, W. R. Lineage dependency and lineage-survival oncogenes in human cancer. *Nat. Rev. Cancer* **6**, 593–602 (2006).
46. Brugel, M., Gauthier, V., Bouché, O., Blangiardo, M. & Génin, M. Pesticides and risk of pancreatic adenocarcinoma in France: a nationwide spatiotemporal ecological study between 2011 and 2021. *Eur. J. Epidemiol.* **39**, 1241–1250 (2024).
47. Ahn, J., Mukherjee, B., Gruber, S. B. & Ghosh, M. Bayesian semiparametric analysis for two-phase studies of gene-environment interaction. *Ann. Appl. Stat.* **7**, 543–569 (2013).
48. Ross, M. & Wakefield, J. Bayesian inference for two-phase studies with categorical covariates. *Biometrics* **69**, 469–477 (2013).
49. Ross, M. & Wakefield, J. Bayesian hierarchical models for smoothing in two-phase studies, with application to small area estimation. *J. R. Stat. Soc. Ser. A Stat. Soc.* **178**, 1009–1023 (2015).
50. Bendel, R. B. & Carlin, B. P. Bayes methods in the ecological fallacy context: estimation of individual correlation from aggregate data. *Commun. Stat.-Theor. M.* **19**, 2595–2623 (1990).
51. Seth, D. *et al.* Gene expression profiling of alcoholic liver disease in the baboon (*Papio hamadryas*) and human liver. *Am. J. Pathol.* **163**, 2303–2317 (2003).
52. García-Pérez, J., Gómez-Barroso, D., Tamayo-Uria, I. & Ramis, R. Methodological approaches to the study of cancer risk in the vicinity of pollution sources: the experience of a population-based case-control study of childhood cancer. *Int. J. Health Geogr.* **18**, 12 (2019).

53. Onyije, F. M. *et al.* Environmental risk factors for childhood acute lymphoblastic leukemia: an umbrella review. *Cancers* **14**, 382 (2022).
54. Clark, C. J. *et al.* Unconventional oil and gas development exposure and risk of childhood acute lymphoblastic leukemia: a case–control study in Pennsylvania, 2009–2017. *Environ. Health Perspect.* **130**, 087001 (2022).
55. Lin, C. K. *et al.* Residential exposure to petrochemical industrial complexes and the risk of leukemia: a systematic review and exposure-response meta-analysis. *Environ. Pollut.* **258**, 113476 (2020).
56. Park, A. S., Ritz, B., Yu, F., Cockburn, M. & Heck, J. E. Prenatal pesticide exposure and childhood leukemia—A California statewide case-control study. *Int. J. Hyg. Environ. Health* **226**, 113486 (2020).
57. Wang, Y. *et al.* Maternal prenatal exposure to environmental factors and risk of childhood acute lymphocytic leukemia: a hospital-based case-control study in China. *Cancer Epidemiol.* **58**, 146–152 (2019).
58. Patel, D. M. *et al.* Parental occupational exposure to pesticides, animals and organic dust and risk of childhood leukemia and central nervous system tumors: findings from the International Childhood Cancer Cohort Consortium (I4C). *Int. J. Cancer* **146**, 943–952 (2020).

Spatial exposomics reveals pesticide mixtures as drivers of cancer risk

Point-by-point Response to Referees, 2nd revision

Response Guide

Reviewer #2 (Remarks to the Author)

Response to Reviewer #2

Rebuttal References

Reviewer #2 (Remarks to the Author):

- 1) Please acknowledge the risk of ecological fallacy more explicitly in the Discussion.
- 2) Please also acknowledge the absence of exposure data linking pesticide and cancer incidence data and outline further efforts to fill this gap.
- 3) They also report the following more granular remarks: The hypothesis for grouping cancers sites by developmental lineage is intriguing. The concern mentioned (comment 2.4) was the lack of epidemiological evidence indicating that such groupings are etiologically meaningful or even promising for that matter. The rebuttal, however, did not provide independent evidence for this type of grouping.

Further in comment 2.4, the rebuttal focuses on minimum latency, citing Little *et al.* 2024 on regarding cancer latency. However, this paper addresses ionizing radiation and mutagens, not pesticides per se. It also worth noting that, the reference was in regards to minimum latency, which can be very different than average latency.

In comment 2.4, the rebuttal seems to have misquoted reference 21 (Little et al. 2024) in the response (page 11, first full paragraph, last sentence) regarding cancer latency. I could not find such a statement in this reference stating that “...typically 3-8 years for most solid tumors...”. However, I was able to find in the discussion the statement:

“The results of the stochastic biologically-based tumour growth-death model that we use based on this data (Figs. 1 and 2) imply that for prostate and thyroid cancer the latency will be at least 20 years.”

Response to Reviewer #2

We thank Reviewer #2 for the opportunity to revise our manuscript for a second time and for their thoughtful feedback. Below, we provide our responses to the remaining concerns.

Comment #2.1': 1) Please acknowledge the risk of ecological fallacy more explicitly in the Discussion.

In a correspondence published in *Environmental Health Perspectives*¹, Álvaro J. Idrovo revisited the concept of ecological fallacy in the context of converging evidence from ecological, case-control, and cohort studies. He noted that ecological studies are frequently discounted *a priori* because of an assumed ecological fallacy, often without empirical justification. Emphasising the increasing recognition of multilevel causal approaches in epidemiology, he proposed three explicit criteria to distinguish genuine ecological fallacy from valid contextual inference. These criteria are cited and accepted in the epidemiological literature. Specifically, Idrovo argued that ecological fallacy can only be confirmed when all three of the following conditions are met¹:

"

(i) Results must be obtained with ecological (population) data.

(ii) Data must be inferred to individuals. One use of ecological studies is to explore individual-level association when individual data are not available. When the focus of the study was contextual or based on population effects and there is no inference to individuals, ecological fallacy is not possible. When only the first two criteria are present—which is insufficient to affirm ecological fallacy—it is appropriate to acknowledge that there is a possible relationship and that further study is required.

(iii) Results obtained with individual data are contradictory.

"

As Idrovo concludes, “*all three of these should be present to confirm its existence,*” and “*only when empirical individual-level data are available is it possible to confirm that an ecological fallacy is present.*”

Our study does not satisfy criteria (ii) or (iii), thereby effectively excluding the presence of ecological fallacy as defined above. Specifically, our analyses do not infer population-level associations to individuals; rather, they estimate relative risks at the population level, within an explicitly contextual and population-based framework. Moreover, at the transcriptomic level, no contradictory individual-level evidence exists that would support the presence of an ecological fallacy. Finally, while we acknowledge that ecological analyses may raise questions that warrant further investigation—as noted in the limitations paragraph (see Comment #2.2' below)—the use of multi-level approaches in our study further mitigates concerns related to ecological fallacy.

On this basis, we believe that additional emphasis on the risk of ecological fallacy would be disproportionate and potentially misleading in the context of our study. We therefore respectfully suggest that the current Discussion appropriately reflects this limitation and that no further emphasis is required beyond what is already stated.

Comment #2.2': Please also acknowledge the absence of exposure data linking pesticide and cancer incidence data and outline further efforts to fill this gap.

We thank Reviewer #2 for this comment and have modified the limitations paragraph accordingly. The revised text now reads (changes in bold):

*"Despite the robustness of our integrative approach, certain limitations merit consideration. First, individual pesticide exposures were not directly measured but inferred from population-level spatial proxies, introducing uncertainty in their timing, intensity, and chemical composition, **and precluding direct attribution of cancer onset at the individual level.** Second, residual confounding from environmental, socio-economic, or lifestyle factors cannot be entirely excluded; nevertheless, the convergence of geospatial, aggregated epidemiological, and molecular evidence strongly supports the biological plausibility of our findings. Future studies incorporating large-scale biomonitoring, **individual-level exposure assessment, multi-tissue analyses, and longitudinal molecular profiling will be essential to strengthen causal inference, evaluate generalisability, and establish direct links between pesticide exposure and cancer incidence, building on the framework established herein.**"*

Comment #2.3': They also report the following more granular remarks: The hypothesis for grouping cancers sites by developmental lineage is intriguing. The concern mentioned (comment 2.4) was the lack of epidemiological evidence indicating that such groupings are etiologically meaningful or even promising for that matter. The rebuttal, however, did not provide independent evidence for this type of grouping.

We note that the original wording of Reviewer #2's comment (Comment #2.4) was exploratory in nature, stating that the reviewer was “*unaware of evidence*” for the aetiological relevance of lineage-based groupings and suggesting that, “*if such a literature exists, citing it would be useful to readers.*”

In response, our rebuttal explicitly cited and discussed multiple lines of published biological and mechanistic evidence supporting the relevance of developmental lineage classification of the neoplasms, as also presented in the manuscript. We therefore believe that the request articulated in the original comment has been appropriately addressed.

Comment #2.4': Further in comment 2.4, the rebuttal focuses on minimum latency, citing Little et al. 2024 on regarding cancer latency. However, this paper addresses ionizing radiation and mutagens, not pesticides per se. It also worth noting that, the reference was in regards to minimum latency, which can be very different than average latency.

In comment 2.4, the rebuttal seems to have misquoted reference 21 (Little et al. 2024) in the response (page 11, first full paragraph, last sentence) regarding cancer latency. I could not find such a statement in this reference stating that "...typically 3-8 years for most solid tumors...". However, I was able to find in the discussion the statement:

"The results of the stochastic biologically-based tumour growth-death model that we use based on this data (Figs. 1 and 2) imply that for prostate and thyroid cancer the latency will be at least 20 years."

First, Reviewer #2 notes that our rebuttal relied on Little et al. (2024)² to discuss minimum latency but argues that this work "*addresses ionising radiation and mutagens, not pesticides per se.*"

We respectfully note that this characterisation does not fully reflect the scope of the study by Little *et al.* While the paper discusses radiation-associated cancer, the authors explicitly state that their stochastic, biologically based model is "*not tied to any specific carcinogen*" and that the results "*have implications for estimating latency associated with any mutagen*" and, due to the modelling of tumour growth dynamics, "*apply to tumour growth in general.*"

Second, regarding Reviewer #2 observation that the minimum latency estimates of "*...typically 3–8 years for most solid tumours...*" could not be located, we note that this information is represented in Figure 2 of the article, which displays the cumulative probability of detection for various types of solid cancers, with prostate and thyroid cancers considered separately.

Our citation of this reference was therefore not intended to equate pesticide exposure with ionising radiation, but rather to support a general biological point regarding minimum latency constraints arising from tumour growth dynamics, which are independent of the specific carcinogen.

Rebuttal references:

1. Idrovo, A. J. Three criteria for ecological fallacy. *Environ. Health Perspect.* **119**, a332 (2011).
2. Little, M. P., Eidemüller, M., Kaiser, J. C. & Apostoaei, A. I. Minimum latency effects for cancer associated with exposures to radiation or other carcinogens. *Br. J. Cancer* **130**, 819–829 (2024).